



# Communicating uncertainties in spatial predictions of grain micronutrient concentration

Christopher Chagumaira[1,2,3], Joseph G. Chimungu[3], Dawd Gashu[4], Patson C. Nalivata[3], Martin R. Broadley[1], Alice E. Milne[2], and R. Murray Lark[1]

[1]School of Biosciences, University of Nottingham, Sutton Bonington Campus, Loughborough, LE12 5RD, United Kingdom
[2]Sustainable Agriculture Sciences Department, Rothamsted Research, Harpenden, Hertfordshire, AL5 2JQ, United Kingdom
[3]Crop and Soil Sciences Department, Bunda College, Lilongwe University of Agriculture and Natural Resources, P O BOX 219, Lilongwe, Malawi
[4]Center for Food Science and Nutrition, Addis Ababa University, Addis Ababa, Ethiopia

**Correspondence:** Christopher Chagumaira (christopher.chagumaira@nottingham.ac.uk; chris.chagumaira@gmail.com)

**Abstract.** The concentration of micronutrients in staple crops varies spatially. Quantitative information about this can help in designing efficient interventions to address micronutrient deficiency. Concentration of a micronutrient in a staple crop can be mapped from limited samples, but the resulting statistical predictions are uncertain. Decision makers must understand this uncertainty to make robust use of spatial information, but this is a challenge due to the difficulties of communicating quanti-
5  tative concepts to a general audience. We proposed strategies to communicate uncertain information and present a systematic evaluation and comparison in the form of maps. We proposed to test five methods to communicate the uncertainty about the conditional mean grain concentration of an essential micronutrient, selenium (Se). Evaluation of the communication methods was done through questionnaire by eliciting stakeholder opinions about the usefulness of the methods of communicating uncertainty. We found significant differences in how participants responded to the different methods. In particular there was a
10  preference for methods based on the probability that concentrations are below or above a nutritionally-significant threshold compared with general measures of uncertainty such as the confidence interval of a prediction. There was no evidence that methods which used pictographs or calibrated verbal phrases to support the interpretation of probabilities made a different impression than probability alone, as judged from the responses to interpretative questions, although these approaches were ranked most highly when participants were asked to put the methods in order of preference.

## 1 Introduction

Micronutrient deficiencies are an important issue in developing countries such as Ethiopia and Malawi. Deficiencies of micronutrients underlay many non-communicable diseases. For example, deficiencies in Se can cause thyroid dysfunction, suppressed immune response and increase disease progression and mortality rates especially in people with already compromised immunity (Fairweather-Tait et al., 2011; Rayman, 2012; Winther et al., 2020).

20    Micronutrients are largely derived from dietary sources, and there is evidence of suboptimal intake of Se below recommended levels in Ethiopia and Malawi (Gashu et al., 2020; Ligowe et al., 2020a). Interventions to improve the dietary intake of Se are





possible. These include agronomic bio-fortification, food diversification and fortification (Broadley et al., 2010; Chilimba et al., 2011; Joy et al., 2019; Ligowe et al., 2020b).

Micronutrient deficiencies and factors that cause them vary spatially (Phiri et al., 2019; Belay et al., 2020; Gashu et al., 2020; Phiri et al.,2020). For example the intake of Se in Ethiopia and Malawi is linked to soil type as well as other factors (Chilimba et al., 2011; Hurst et al., 2013; Joy et al., 2015). Belay et al. (2020) showed that the risk of Se deficiency is widespread and spatially dependent across Ethiopia. So spatial information (e.g. on grain micronutrients) can be used to design more efficient interventions to address micronutrient deficiency.

Soil and crops cannot be sampled everywhere and measurements can only be made directly at a few locations. Using statistical models, interpolations at unsampled locations can be made but the predictions are uncertain. Predictions are subject to uncertainty because of spatial variability resulting from multiple factors operating at different scales (Lark et al., 2014) including the short-range variation described by the nugget variance (Lark et al., 2016). When using spatial information, it is therefore important to report this uncertainty and make sure that decision-makers understand it in order to make informed decisions.

In geostatistical prediction, the uncertainty of a predicted value is quantified directly by the kriging variance, the mean squared error of the prediction. The prediction is a linear combination of the data which is optimal in the sense of minimizing the kriging variance, given a variogram function which models the spatial dependence of the variable of interest. The kriging variance depends, given the variogram, on the spatial distribution of observations. Assuming normal prediction errors, the kriging variance can be used to compute a confidence interval for the prediction. It is therefore possible to represent the uncertainty in a map of micronutrient concentrations in grain by a corresponding map which shows the kriging variance, or by the upper and lower bounds of the confidence interval, which can also be mapped.

Other approaches can be taken to communicate the uncertainty in a prediction when the prediction is to be interpreted relative to some threshold value of the mapped variable (e.g. a threshold concentration below which typical intake of grain does not provide adequate intake of a nutrient). While the predicted value may lie above the threshold, because the prediction is uncertain it is possible that the true value is actually below the threshold. This probability, conditional on the data and on the geostatistical model, can be obtained in various ways. A common geostatistical approach is to use indicator kriging (e.g. Webster and Oliver, 2007).

The quantification of uncertainty is generally straightforward, but the communication of this uncertainty to a range of users of information is less so. As Milne et al. (2015) found, the success of a method to present uncertainty may depend on the subject matter and on the background of the interpreter. The probability that the true value lies below a threshold might not be easily interpreted by the policy maker or manager who needs to make a decision based on a map. Probability is often not easily-interpreted by a range of end-users of information (Spiegelhalter et al., 2011), and for this reason, in addition to the 'raw' probability, verbal interpretations of probability based on 'calibrated phrases' (e.g. 'unlikely') have been proposed — e.g. the Intergovernmental Panel for Climate Change (IPCC) scale due to (Mastrandrea et al., 2010). Pictographs may also be used to communicate probabilities by enabling the interpreter to visualize them as proportions (e.g. Spiegelhalter et al., 2011).





Statistical predictions can be used to support decision making to identify areas of sufficiency or insufficiency. A simple decision model could be based on a threshold value, of a variable with the aim that the user should act if the variable of interest falls below or exceeds the threshold. In our study we chose a threshold of 38 µg kg$^{-1}$, based on the assumption that a mean daily intake of 330g of grain flour should provide a third of the daily estimated average requirements (EAR) of Se for an adult

woman. The EAR is a commonly-used measure of intake when assessing nutritional status and planning intervention.

In this study we propose methods to communicate uncertainty in mapped concentrations of micronutrients in grain using Se as a case-study. These methods are based on the kriging variance or on the probability that concentration falls below a nutritionally-significant threshold. Maps using these methods, and based on real data collected in Ethiopia and Malawi were presented to panels of stakeholders in those countries, and their experience of using the maps, and their evaluation of the

different methods were recorded using questionnaires.

## 2 Materials and Methods

This study was conducted in Ethiopia and Malawi. Ethiopia is located in the horn of Africa (9.1450° N, 40.4897° E), while Malawi is in southern Africa (13.2543° S; 34.3015° E). Primarily, these are research sites for the GeoNutrition project (http://www.geonutrition.com/) to inform strategies on addressing micronutrient deficiencies commonly referred to as 'hid-

den hunger'. We proposed to test five methods to communicate the uncertainty about predictions of Se concentration in grain (see Section 2.1).

In order to determine how best to communicate the uncertainty in our predictions, we recruited participants to evaluate our five candidate methods at two workshops held in Lilongwe, Malawi (November 2019) and Addis Ababa, Ethiopia (January 2020). Each method was presented on a poster, with the same format, consisting of (1) predicted nutrient concentration (con-

ditional mean) in map form, and (2) a map communicating the uncertainty about the predictions. Examples of the posters are shown in Figs. S1 to S5, in the supplementary materials. Formal evaluations were done through a structured questionnaire that participants completed during the workshops. Ethical approval to conduct this study was granted by the University of Notting-ham School of Sociology and Social Policy Research Ethics Committees (BIO-1920-004 for Malawi, and BIO-1920-007 for Ethiopia).

### 2.1 Test Methods

#### 2.1.1 Statistical modelling and spatial prediction of grain Se concentration

Field sampling in Amhara, Ethiopia, was previously conducted to support spatial prediction of Se concentration in grain crops including the staple crops teff (*Eragrostis tef* (Zucc.) Trotter) and wheat (*Triticum aestivum* L.) (Gashu et al., 2020). The sample frame was defined with reference to the Africa Soil Information Service map of croplands in Amhara region (AfSIS,

2015) so that all sample sites were expected to have a crop or to be near a cropped site. The sample points were selected to give good spatial spread across the sample frame, and to be spatially balanced. This procedure was implemented in the





BalancedSampling library for the R platform (R Core Team, 2017, Grafström and Lisic, 2016). A total of 25 additional sample sites, closely paired with one of those selected as described above, were added to the sample design to support the estimation of the parameters of the spatial linear mixed model (Gashu et al., 2020). In total, 455 sampling points was obtained, including 136 and 113 locations where teff and wheat were sampled, respectively.

In Malawi, the objective of field sampling was to support spatial prediction of Se concentration in maize (*Zea mays* L), the staple crop. The location of sample points were obtained with the spcosa package for the R platform (Walvoort et al., 2010). This finds sample points which give good spatial coverage of a sample domain, and can incorporate the location of fixed prior points. We had 820 prior points from the 2015–16 micronutrient survey of Malawi (Phiri et al., 2019), and added a further 890 spatial coverage points with spcosa. Of these 1710 sites, 190 were selected at random for a duplicate 'close pair' sample to support spatial modelling–10% of the total samples following Lark and Marchant (2018).

We first undertook exploratory data analysis, using simple summary statistics and plots, notably Q-Q plots to check whether we needed to transform the data to make the assumption of normality reasonable. In order to check for any spatial trends we plotted classified post-plots which show the spatial location of data and use symbols to indicate quantiles. We found no evidence of spatial trend in the Malawi data. The data were very skewed and we transformed the data to logarithms, to make the assumption of normality plausible. However, for the Amhara dataset, we observed a spatial trend. Exploratory analysis indicated that a linear trend model in the spatial coordinates accounted for this, and exploration of the residuals from the trend indicated that a transformation to logarithm was necessary.

After the exploratory data analysis, we used ordinary kriging to obtain the kriging prediction (conditional mean) and kriging variance of grain Se concentration in Malawi dataset for every prediction location. However, for the Amhara data we used universal kriging which also makes predictions at unsampled locations, $\mathbf{x}_0$, by a weighted linear combination of available sample data designed to minimise prediction error whilst filtering the trend (Webster and Oliver, 2007). The variance parameters for both Amhara and Malawi datasets were estimated by residual maximum likelihood (Diggle and Ribeiro, 2010) with the likfit procedure for the R platform.

We used indicator kriging to obtain the conditional probability that grain Se concentration at the unsampled location exceeds the threshold value, 38 μg kg$^{-1}$. Indicator kriging predictions are made by ordinary kriging of an indicator variable created by a transformation of the data on a variable of interest, $z$, to an indicator variable $w$, given a threshold value of interest, $z_{\mathrm{T}}$. The indicator variable at location $\mathbf{x}$ takes the value 0 if $z(\mathbf{x}) \leq z_{\mathrm{T}}$ and 1 otherwise. The estimate of the indicator variable at some location $\mathbf{x}_0$ can be interpreted as the conditional probability that $z(\mathbf{x}_0) \leq z_{\mathrm{T}}$ (Webster and Oliver, 2007).

### 2.1.2 Kriging Variance

In statistical predictions, some unknown quantity (e.g. grain Se concentration at a location) has a prediction distribution conditional on data and a statistical model. The mean of the prediction distribution (conditional mean) is most commonly treated as the predicted value. The best linear unbiased prediction, such as the kriging prediction, is the conditional mean from the linear mixed model. The variance of the conditional distribution is the prediction error variance, known as the kriging variance in geostatistics. It is evaluated at every prediction location, it can be presented as a map alongside the mapped predictions. At





each kriged estimate, of grain Se concentration, expected squared error of the prediction is defined as,

$$\sigma_K^2 = \mathrm{E}[\{z(\mathbf{x}_0) - \tilde{Z}(\mathbf{x}_0)\}^2], \tag{1}$$

where $z(\mathbf{x}_0)$ is measured data and $\tilde{Z}(\mathbf{x}_0)$ is the prediction by ordinary kriging or universal kriging. The map of kriging variance is a summary of the uncertainty about our predictions in the study area and shows areas that need further sampling to resolve

uncertainty for decision making. In ordinary kriging, the kriging variance has smaller values near the sample location and so reflects the distribution of sampling points. For universal kriging, the kriging variance is smallest near sample location where the values of covariates are close to their respective mean. To investigate the utility of the kriging variance as a method to communicate uncertainty, one poster showed a map of conditional mean of Se concentration in grain (Section 2.1.1), with a map of kriging variance (see Table 1).

### 130  2.1.3  Confidence Intervals

We computed cross-validation predictions from our geostatistical model and exploratory analysis of the kriging errors, $\{z(\mathbf{x}_0) - \tilde{Z}(\mathbf{x}_0)\}$, showed that these can be regarded as a normal random variable. Because the kriging predictor is unbiased the mean of the errors is zero and their standard deviation is equal to kriging standard deviation $\sigma_K(\mathbf{x}_0)$. On this basis we computed a 95% confidence interval at each prediction location as $\tilde{Z}(\mathbf{x}_0) \pm 1.96\sigma_K(\mathbf{x}_0)$. One poster showed a map of conditional mean of

Se concentration in grain plus the lower and upper bounds of the 95% confidence intervals mapped separately to communicate the uncertainty (see Table 1).

### 2.1.4  Conditional Probability

Using indicator kriging allows us to quantify uncertainty of the prediction in terms of the probability that the true value exceeds or lies below the threshold. This is a conditional probability, conditional on the data and indicator variogram. The probability

provides a basis for decisions on interventions given the threshold value. For example, if the conditional probability that grain Se is below the threshold is very large then a decision might be made to promote an intervention such as dietary supplementation or agronomic biofortification.

Probability can be presented in a number of different ways, at the first instance on a raw probability scale, from 0 to 1 or 0 to 100%. However, raw probabilities are not very useful to non-specialists as they are often misinterpreted (Spiegelhalter et

al., 2011). Given this shortfall, the Intergovernmental Panel on Climate Change (IPCC) (Mastrandrea et al., 2010) introduced a verbal scale for communicating probabilistic information from uncertain results using 'calibrated' verbal phrases. For example, an event with probability <1% will be described as '*Exceptionally unlikely*' and an event with probability in the interval 90–99% is described as '*Very likely*'. However, the scale is not always interpreted consistently among different individuals. Budescu et al. (2009) observed a tendency to "regressive" interpretation in which large or small probabilities are interpreted as close

to 50%. Therefore, we followed Lark et al. (2014) in supplementing the 'calibrated' verbal phrases with the definition of the probability range.





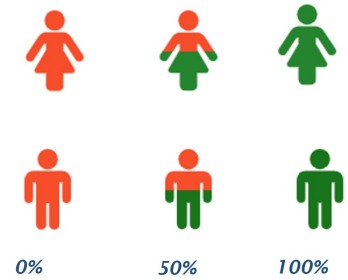

**Figure 1.** Use of pictographs reporting a probability of an event exceeding a threshold.

**Table 1.** The designated poster number for each method of communicating uncertain information.

| Poster | Method of Communication |
|---|---|
| Poster 1 | Confidence Interval |
| Poster 2 | IPCC Verbal Scale |
| Poster 3 | Kriging Variance |
| Poster 4a | Raw Probability |
| Poster 4b | Raw Probability plus Pictograph |

Graphics, such as pictographs, can be used to report the probability of an event exceeding a threshold. Graphics can be tailored for the target audience and can help those with low numeracy. Zikmund et al. (2008) showed that pictographs significantly improved people's understanding of disease risks compared with other graphics. However, Spiegelhalter et al. (2011) suggested that graphics such as pictographs can be misinterpreted particularly by people of low numeracy. Therefore, in this study we proposed to combine raw probabilities and graphics to communicate uncertainty to address these set backs. In the exercise we did it by showing the probability map and the pictograph for locations of interest. We used pictographs to report probability of grain Se concentration exceeding the threshold value, as shown in Fig. 1.

Therefore, we presented three posters, each showing a map of conditional mean of Se concentration in grain (Section 2.1.1.), plus probability presented as (1) raw probability scale, (2) IPCC verbal scale and (3) raw probability scale plus pictographs, communicating the uncertainty (see Table 1).

### 2.2 Format of the exercise

We wanted to elicit stakeholder opinions about the usefulness of the communication methods presented as posters described in Section 2.1. We invited participants working in the following sectors: agriculture, nutrition and health, NGOs, universities and government departments from Ethiopia, Malawi and in the wider GeoNutrition project sites. In Ethiopia, through a focal person in the GeoNutrition project, we recruited participants who fitted in the above criterion and these were mainly local professionals. In Malawi, through focal persons at Lilongwe University of Agriculture and Natural Resources, we invited





**Table 2.** The composition of participants during the meetings in Ethiopia and Malawi.

| Meeting/ | Number of Participants | | | Total |
|---|---|---|---|---|
| Country | Agronomist | Soil Scientist | Nutritionist /Health Practitioner | |
| *Ethiopia Meeting* | | | | |
| Ethiopia | 6 | 13 | 17 | 36 |
| *Malawi Meeting* | | | | |
| Ethiopia | - | 1 | 1 | 2 |
| Malawi | 6 | 5 | 4 | 15 |
| Pakistan | - | 2 | - | 2 |
| Zambia | - | 2 | - | 2 |
| Zimbabwe | - | 2 | 2 | 4 |
| Total | 12 | 25 | 24 | 61 |

participants who fitted the above criterion. Many of the participants were already engaged with the GeoNutrition project. In total we had 61 participants, 36 in the Ethiopia meeting and 25 in the Malawi meeting (see Table 2). We asked our participants

to assign themselves into one of the three professional groups (1) 'agronomist', (2) 'nutritionist/health practitioner' and (3) 'soil scientist'. We then asked them to record their level of mathematical education and level of use of statistics or mathematics in their job role.

Evaluation of the communication methods was done through questionnaire, as shown in Table 3, but without putting the participants in a situation were they felt they where being tested on their mathematical skills and understanding. The first part

of the questionnaire was an interpretative task, Questions 1 to 3 (Q1 to Q3). We presented them with true statements about the confidence in the information presented on the maps, at different locations ($x$, $y$, and $z$). We asked whether the communication of uncertainty was clear. Then we had the decision-focused task, Q4, where we asked whether each poster (prediction plus uncertainty) provided adequate information to support a given decision. We then had reflective tasks Q5 and Q6. In Q5, we asked whether in each case the uncertainty about grain Se concentration was straightforward to interpret. We asked if the

method of communication helped them understand uncertainty in the predictions in Q6. At the end of the questionnaire, we wanted the participants to assess the methods (Q7) by ranking the posters in order of their effectiveness at communicating uncertainty in the predictions.

In each workshop, we started out with an introductory talk to explain the objectives of the exercise. During the talk, we also explained the structure of the questionnaire and how we expected the participants to complete it. After being handed the

questionnaires, the participants were directed into a room with the five methods displayed on A0 sized posters. Participants



**Table 3.** The list of questions used to elicit stakeholder opinions about the usefulness of the communication methods presented as posters in the workshops in Ethiopia and Malawi.

| Question | | Response |
|---|---|---|
| Question 1 (Q1) | Is it clear from the poster,that this statement is true? "Our confidence that grain Se concentration exceeds 38 µg kg$^{-1}$ is greater at $x$ than at $z$" | (1) Not clear, (2) Took a while, (3) Can be misinterpreted, (4) More information needed, (5) Message clear |
| Question 2 (Q2) | Is it clear from the poster, that this statement is true? "Our confidence that grain Se concentration does not exceed 38 µg kg$^{-1}$ is greater at $z$ than at $y$" | (1) Not clear, (2) Took a while, (3) Can be misinterpreted, (4) More information needed, (5) Message clear |
| Question 3 (Q3) | Is it clear from the poster,that this statement is true? "Our confidence that grain Se concentration does not exceed 38 µg kg$^{-1}$ is greater at $y$ than at $x$" | (1) Not clear, (2) Took a while, (3) Can be misinterpreted, (4) More information needed, (5) Message clear |
| Question 4 (Q4) | Does the poster provide adequate information for you to determine how likely it is that an intervention programme is needed at any given location? | (1) Inadequate information, (2) Adequate information, (3) More than what I wanted |
| Question 5 (Q5) | Is the way this poster communicates the uncertainty about grain Se concentration straightforward to interpret? | (1) Not clear, (2) Took a while, (3) Can be misinterpreted, (4) More information needed, (5) Message clear |
| Question 6 (Q6) | Do you think that the poster helped you understand the uncertainty in the predictions? | (1) Yes, (2) No |
| Question 7 (Q7) | Comparing all methods please rank the posters in order of their effectiveness, in your experience, at communicating uncertainty in the predictions | Rank 1 being most effective, Rank 5 the least. |

visited each poster in a randomised order to avoid any bias resulting from carry-over effects from one poster to another. For example, if participants found a particular method easier to interpret, this might help them understand the next poster that they examined. Participants were not allowed to speak to one another when they were completing their questionnaires to avoid bias. When completing the last two questions on the questionnaire, participants were allowed to revisit the posters without following





the randomised order to revise their answers. A non-specialist facilitator was stationed at the poster, to check that participants were on the correct pages on the colour coded questionnaire, to check that all questions were completed and to help with any problems (e.g. translating language).

## 2.3    Data Analysis

We presented our results for Q1 to Q6 as contingency tables, where the selected responses are in rows (of which there are $n_{\mathrm{r}}$)
and the columns (of which there are $n_{\mathrm{c}}$) are the posters (i.e. methods of communication) separated either between location of meeting (Ethiopia or Malawi) or between professional group (agronomists or soil scientist or nutritionist/health practitioner) of the respondent. The 'full table' illustrated in Fig. 2 is an example of this. Analysis of the contingency table allows us to test the null hypothesis of random association of the responses with the factor in columns (i.e. that the proportion of participants indicating a particular response to the question is independent of the poster which they are considering).

The null hypothesis for a contingency table is equivalent to an additive log-linear model of the table under which the expected number of responses in cell $[i,j]$, $e_{i,j}$, is the product of the row and column totals ($n_i$ and $n_j$) divided by the total number of responses, $N$. An alternative log-linear model, the so-called 'saturated' model for the table, has an extra $(n_{\mathrm{r}} - 1) \times (n_{\mathrm{c}} - 1)$ terms which allows an interaction between rows and columns of the table such that the proportions of different responses may differ among all the posters.

The evidence for the saturated model, as a better model for the data than the additive model, is provided by the likelihood ratio statistic or deviance for the two models, $L$, where

$$L = 2 \sum_{i=1} \sum_{j=1} o_{i,j} \log \frac{o_{i,j}}{e_{i,j}}. \tag{2}$$

Under the null hypothesis of random association between the rows and columns of the table, $L$ has an approximate $\chi^2$ distribution with $(n_r - 1) \times (n_c - 1)$ degrees of freedom (Christensen, 1997; Lawal, 2014). We fitted the log-linear models using
the loglm function from the MASS package in the R platform (Venables and Ripley, 2002).

    A full table, such as the one shown in Fig. 2, may be hard to interpret. It is possible to partition the table, and its deviance statistic and degrees of freedom, into components corresponding to pooled tables and subtables of the full table. This is illustrated in Fig. 2. Here the full table is partitioned into a subtable for responses from Malawi and another subtable for responses from Ethiopia. A pooled table, in which the responses pooled over all posters in Malawi were compared with the responses
similarly pooled from Ethiopia, completes the partition. As shown in Fig. 2, the deviance statistics for these three tables, and their degrees of freedom, sum to the deviance and degrees of freedom for the full table. In this case we could conclude whether there are differences in the responses between the two locations (if not, then we might pool the responses for any poster at the two locations), and whether there are differences in responses to the posters at each location in turn. As described below, we used this approach to evaluate whether there were differences between the two locations. We also used it to examine evidence
for differences in the responses for professional groups. Having done this, we then analysed either pooled tables or separate subtables (e.g. for responses in Ethiopia and responses in Malawi) to examine a *priori* contrasts between particular posters and groups of posters.

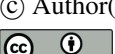



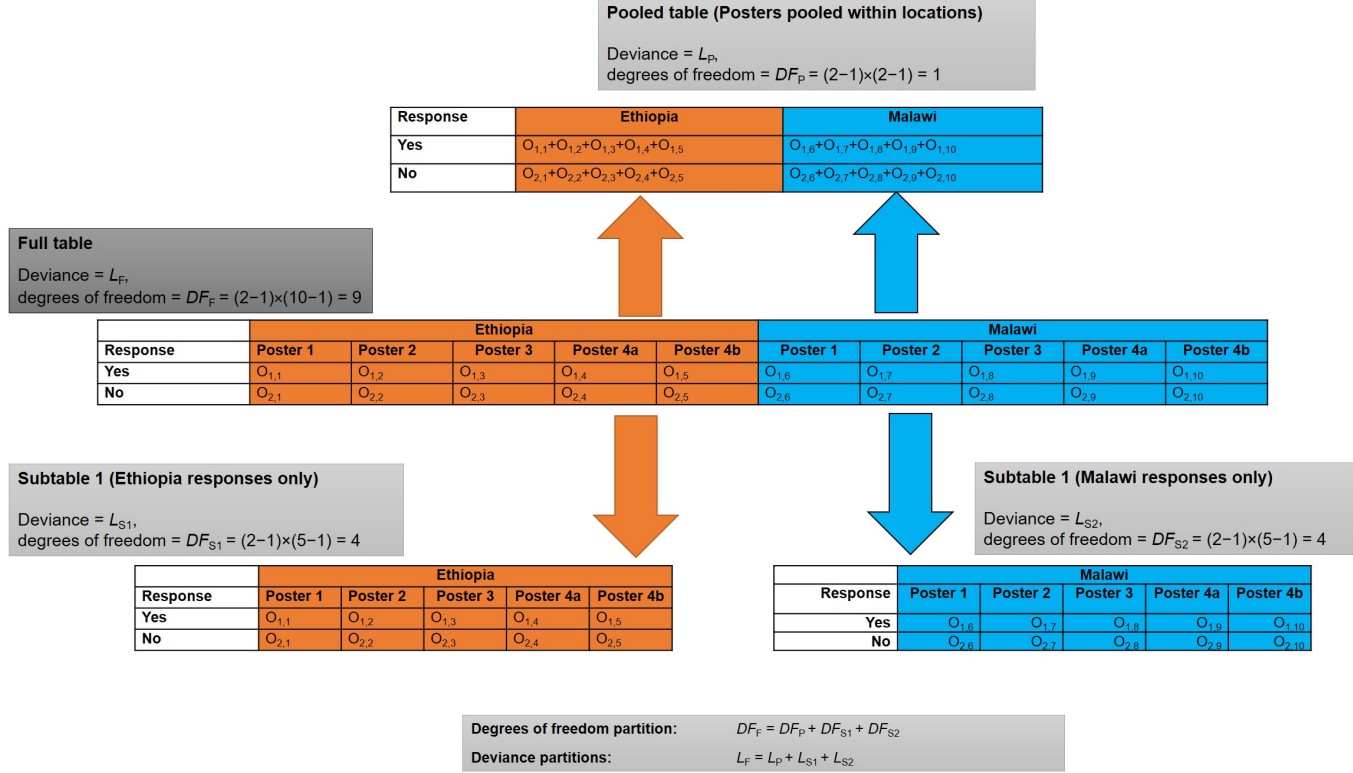

**Figure 2.** An illustration how the log-likelihood ratio can be partitioned into sub-tables and pooled tables.

Our primary interest is whether there are differences between responses recorded by our participants depending on the method of communicating uncertainty. However, it was first necessary to consider whether there was evidence for differences

in the responses between the between the two sets of respondents at different locations. Such differences might arise because of differences in the composition of the groups (Table 2), differences between the examples presented (a map from Amhara Region in Ethiopia, a map of Malawi in Malawi), differences between the contexts (in Ethiopia many were local professionals recruited for the exercise, in Malawi many of the participants were already engaged with the GeoNutrition project), and the possibility of unconscious changes in how the second meeting, in Ethiopia, was conducted (adapting from the experience

of conducting the exercise in Malawi). Because our participants are drawn from different professional groups, we thought this might affect their responses, and if so this would also be of interest because it would suggest that people from different professional backgrounds find some methods better than others.

For this reason, we first tested whether there were differences in the overall responses between the location of meetings, using a contingency table in which the responses to different posters by people from different professional groups are pooled within

the two meeting locations. This gives us a five (responses) by two (locations) contingency table, with 4 degrees of freedom for each poster ( Q1 to Q3 and Q5) or a three (responses) by two (locations) contingency table, with 2 degrees of freedom (Q4)





or a two (responses) by two (locations) contingency table, with 1 degrees of freedom (Q6). We next tested whether there were differences in the overall responses between the different professional groups, using a contingency table in which the responses to different posters were pooled within each of those groups.

For some questions, there were differences in the responses between location of meeting. But for no questions was there any evidence to reject the null hypothesis of random association between responses and the professional group of the participants. We therefore proceeded to consider a set of prior hypotheses about differences in the responses between posters, and the methods which they employed to communicate uncertain information, based either on a partition of the separate subtables for each location (where the locations differed) or of a table in which the responses from the different locations were pooled.

The first hypothesis which we considered is that participants would respond differently to a threshold based approach to uncertainty (in which the poster presents the probability that the Se concentration in grain at an unsampled site falls below or above a threshold, posters 2, 4a and 4b), than they would to a general measure of uncertainty (the kriging variance, poster 3, or the confidence interval for the prediction, poster 1). We call this hypothesis $H^1$, and the evidence against the corresponding null hypothesis $H_0^1$, was evaluated by the deviance for the subtable in which the responses to posters 2, 4a and 4b were pooled

in one column (threshold based) and the responses to posters 1 and 3 were pooled in a second.

The second hypothesis that we considered, $H^2$ was that the respondents' views on the posters that used kriging variance would differ from their views on the posters that used confidence intervals. The evidence against the corresponding null hypothesis, $H_0^2$, was tested by the subtable comprising the responses to poster 1 in one column and the responses to poster 3 in a second.

The deviances for the tables testing null hypotheses $H_0^1$ and $H_0^2$ are two components of the deviance for the overall table (be this pooled over locations or a subtable for one location). The remaining deviance component is for a subtable with all the separate responses to threshold based methods. This can be partitioned into two further components, which address our two remaining hypotheses.

The first of these, hypothesis $H^3$, was that respondents would have different opinions about poster 4a (raw probability

values) than the posters (4b, 2) in which guides to the interpretation of the probability are given (pictographs, or partition of the probability into intervals corresponding to the calibrated phrases of the IPCC scheme). The null hypothesis $H_0^3$ is tested by the deviance of a table in which one column comprises responses to poster 4a, and the second contains pooled responses to posters 4b and 2.

The final hypothesis, $H^4$, was that respondents would have different opinions on the poster which used the calibrated phrases

of IPCC (poster 2) and the rather different approach of poster 4b with pictographs imposed on a map of probabilities.

The approaches above were applied for all of questions Q1 to Q6.

We tabulated responses for Q7, with ranks as the rows, and posters as the columns. Participants were asked to rank the preferred poster first, but we reversed this for the analysis, giving a rank of 5 to the most preferred poster and of 1 to the least. We considered only those responses where a complete ranking was provided by the respondent. The mean rank was calculated

for each poster, and this was done over all respondents, and then separately for locations and for professional groups.





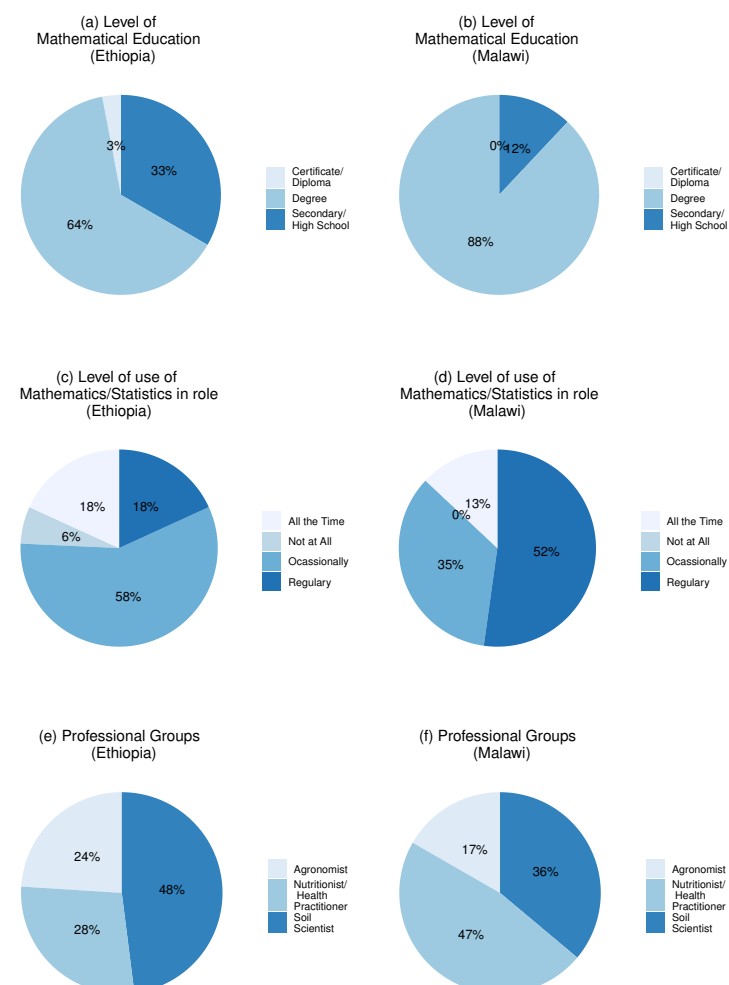

**Figure 3.** The percentage of participants by level of mathematical education and use of mathematics or statistics in their role.

For a set of rankings of $k$ items, under a null hypothesis of random ranking, the expected mean rank for each item is $(k+1)/2$. The evidence against this null hypothesis can be measured by the statistic:

$$\frac{12n}{k(k+1)} \sum_{i=1}^{k} \left\{ \bar{r}_i - \frac{k+1}{2} \right\}^2, \tag{3}$$

where $\bar{r}_i$ is the mean rank of the $i$th item, and a total of $n$ rankings comprise the data. Under the null hypothesis this statistic is
distributed as $\chi^2(k-1)$ (Marden, 1995).





## 3 Results

In the Ethiopia meeting, we had less participants (64%) who had studied mathematics and statistics up to degree level and above, than in the Malawi meeting (88%), see Fig. 3. We had more participants using statistics or mathematics regularly in their job in the Malawi meeting (52%) than in the Ethiopian meeting (18%). Most of the participants in the Ethiopian meeting (58%)

occasionally use mathematics or statistics in their job role. There were more soil scientists (48%) at the meeting in Malawi than agronomists and nutritionists/health practitioners. Whilst in Ethiopia, there were more nutritionists/health practitioners (47%) compared to the other professional groups.

### 3.1 Interpretative Tasks

Table 4 shows the full table for responses over both locations and all posters to question Q1. The responses pooled for both

meeting locations are shown in Table 5. There is strong evidence for differences among the columns of the full table ($P<0.001$), and strong evidence ($P<0.001$) against the null hypothesis of random association between posters and responses pooled within locations and responses (Table 6). However, there was no evidence to reject the null hypothesis of random association between posters and responses pooled within professional groups. On this basis further analysis of responses to posters was based on the separate subtables for the Ethiopia and Malawi meeting locations. Similar results were obtained for Q2 and Q3 as shown

in Tables 7 and 8, respectively.

For question Q2, while there is evidence for a difference in responses between the two meeting locations, there is no evidence, either for the responses from Ethiopia or from Malawi, to reject the null hypothesis for any of the focussed questions about differences between posters, see Table 7. For Q3, however, there is evidence for a difference in the responses for the threshold based methods and the general methods in the responses from Ethiopia ($P=0.009$) and from Malawi ($P=0.02$) (see Table 8).

Fig. 4 shows the responses to Q1 for the separate posters for each subtable. Threshold based methods were found to be clearer by larger proportion of the participants. In both countries, there was a marked difference between poster 1 (confidence intervals) and the rest, with a much smaller proportion of respondents selecting the response 'Message clear'. In Malawi, a large proportion of respondents selected 'Not clear' as their response for this poster. The figures which summarize responses for Q2 and Q3 are shown in the supplementary information (Figs. S9 and S10).

### 3.2 Decision-focused task

There is no evidence for differences among the columns of the full table ($P=0.11$) and strong evidence ($P=0.01$) against the null hypothesis of random association between posters and responses pooled within locations and responses, for Q4 (Table 9). However, there was no evidence to reject the null hypothesis of random association between posters and responses pooled within professional groups. Therefore, further analysis of responses to posters was based on the separate subtables for the

Ethiopia and Malawi meeting locations.





**Table 4.** The full contingency table showing how many individuals selected a given response to Q1, interpretive task. The table is presented according to location of meeting and method of communication. The figures in parentheses are the expected numbers, $e_{i,j}$ the product of the row and column totals ($n_i$ and $n_j$) divided by the total number of responses, $N$.

| Response | Ethiopia | | | | | Malawi | | | | |
|---|---|---|---|---|---|---|---|---|---|---|
| | Poster 1 | Poster 2 | Poster 3 | Poster 4a | Poster 4b | Poster 1 | Poster 2 | Poster 3 | Poster 4a | Poster 4b |
| Not clear | 1(1) | 0(1) | 4(1) | 1(1) | 0(1) | 8(3) | 1(3) | 5(3) | 2(3) | 1(3) |
| Took a while | 9(7) | 8(6) | 6(6) | 6(7) | 4(7) | 0(1) | 1(1) | 3(1) | 2(1) | 1(1) |
| Can be misinterpreted | 5(4) | 4(4) | 3(4) | 5(4) | 3(4) | 6(2) | 1(2) | 3(2) | 0(2) | 0(2) |
| More information needed | 7(3) | 2(3) | 2(3) | 2(3) | 3(3) | 2(2) | 0(2) | 3(2) | 3(2) | 0(2) |
| Message clear | 13(20) | 20(19) | 19(19) | 22(21) | 26(21) | 8(16) | 22(16) | 11(16) | 18(16) | 22(16) |

**Table 5.** A subtable showing how many individuals selected a given response to Q1 when columns are pooled within location of meeting.

| Response | Ethiopia | Malawi |
|---|---|---|
| Not clear | 6 | 17 |
| Took a while | 33 | 7 |
| Can be misinterpreted | 20 | 8 |
| More information needed | 16 | 8 |
| Message clear | 100 | 81 |

For Q4, we have no evidence to reject the null hypothesis of random association between poster and response for any of our set of four focussed hypotheses in Ethiopia. In Malawi, however, there is evidence ($P$=0.03) to reject the $H_0^1$, and not for the other focussed hypotheses.

Fig. 5 shows the responses to Q4 for the separate posters for each subtable graphically. The larger proportion of the par-
ticipants found threshold based methods to provide adequate information for decision making. In Ethiopia, poster 3 (kriging variance) was different from all other posters, with a large proportion of respondents selecting 'Inadequate information'.

### 3.3 Reflective task

There is no evidence for differences among the columns of the full table ($P$=0.26) for Q5 (Table 10). Also, there is no evidence ($P$=0.63) against the null hypothesis of random association between posters and responses pooled within locations. Table 11
shows that there is strong evidence for differences among the columns of the full table ($P$=0.001) for Q6. However, the evidence is marginal ($P$=0.05) against the null hypothesis of random association between posters and responses pooled within locations and responses. However, there was no evidence to reject the null hypothesis of random association between posters




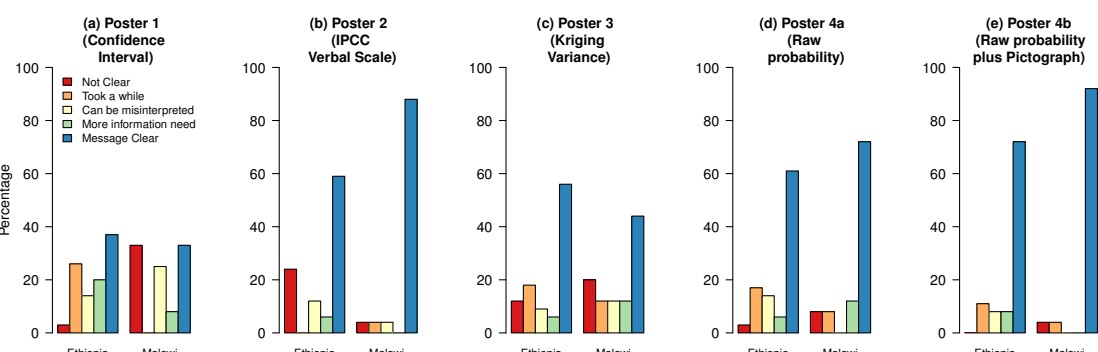

**Figure 4.** Bar charts showing how participants when pooled within location of meeting responded to the interpretive task, Question 1.

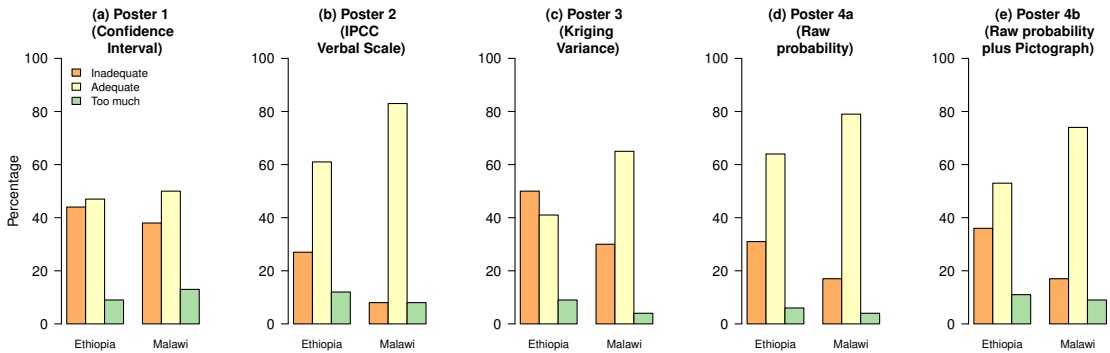

**Figure 5.** Bar charts showing how participants, when pooled within location of meeting, responded to whether a method provided adequate information or not, Question 4.

and responses pooled within professional groups for both Q5 and Q6. On this basis further analysis of responses to posters was based on pooled counts for the Ethiopia and Malawi meetings. The responses for Q5 are shown in Table 12.

As shown in Table 10, we have evidence ($P$=0.02) to reject the null hypothesis of contrasting the threshold based methods with the general uncertainty measures for Q5. For Q6, there is evidence for a difference in the responses for the threshold based methods and the general methods ($P$<0.001). However, we have no evidence for the second, third and forth focussed hypotheses in both Q5 and Q6.

     Fig. 6 shows the responses to Q5 for the separate posters for pooled counts graphically. We can see that a there is larger
proportion of respondents selecting the response 'Message clear' on threshold based methods, Posters 2 (IPCC verbal scale), 4a (raw probability) and 4b(raw probability plus pictograph), than on general based. We also see more people selected the response 'Not clear' on posters 1 (confidence interval) and 3 (kriging variance), the general based methods. Fig. 7 shows how





**Table 6.** Analysis of Q1 according to location of meeting, professional group and methods the latter tested on separate location subtables.

| | Specified null hypothesis[†] | Deviance $(L^2)$ | Degrees of Freedom | $P^*$ |
|---|---|---|---|---|
| *Full contingency table analysis* | | | | |
| Full table | | 93.33 | 36 | <0.001 |
| Pooled within location of meeting | | 22.83 | 4 | <0.001 |
| Pooled within professional group | | 11.71 | 8 | 0.16 |
| *Subtable- Ethiopia meeting* | | | | |
| Poster effects | | 21.78 | 16 | 0.15 |
| Threshold based vs General | $H_0^1$ | 9.61 | 4 | 0.05 |
| Within general | $H_0^2$ | 7.10 | 4 | 0.13 |
| Within threshold based | | 5.07 | 8 | 0.75 |
| Poster 4a vs Guided | $H_0^3$ | 2.64 | 4 | 0.62 |
| Poster 4b vs Poster 2 | $H_0^4$ | 2.43 | 4 | 0.66 |
| *Subtable- Malawi meeting* | | | | |
| Poster Effects | | 48.72 | 16 | <0.001 |
| Threshold based vs General | $H_0^1$ | 31.95 | 4 | <0.001 |
| Within general | $H_0^2$ | 6.53 | 4 | 0.16 |
| Within threshold based | | 10.24 | 8 | 0.25 |
| Poster 4a vs Guided | $H_0^3$ | 8.87 | 4 | 0.06 |
| Poster 4b vs Poster2 | $H_0^4$ | 1.37 | 4 | 0.85 |

[†] Each row of this table presents a test of a null hypothesis of random association between the rows and columns of a contingency table, but the four highlighted here correspond to the prior hypotheses about differences among posters which are of primary interest.

$^*$ Probability of obtaining a deviance statistic this large or larger if the null hypothesis of random association of the rows and columns of the table hold.

participants responded to Q6. There was a marked difference between poster 3 (kriging variance) and the rest, with a much larger proportion of respondents selecting the response 'No'.

**3.4 Assessment of the method**

For Q7, firstly we computed the mean ranks for all the participants and measured the evidence against the null hypothesis of random ranking using the Equation 3. Table 13 shows that there is strong evidence ($P$=0.002) against the null hypothesis of random ranking

Secondly, we computed mean ranks for each location of the meeting. After the test, we found no evidence ($P$=0.12) against
the null hypothesis in Ethiopia. However, in the Malawi meeting there was strong evidence ($P$=0.001).





**Table 7.** Analysis of Q2 according to location of meeting, professional group and methods the latter tested on separate location subtables.

| | Specified null hypothesis[†] | Deviance ($L^2$) | Degrees of Freedom | $P^*$ |
|---|---|---|---|---|
| *Full contingency table analysis* | | | | |
| Full table | | 60.66 | 36 | 0.01 |
| Pooled within location of meeting | | 24.42 | 4 | <0.001 |
| Pooled within professional group | | 14.95 | 8 | 0.06 |
| *Subtable- Ethiopia meeting* | | | | |
| Poster effects | | 16.21 | 16 | 0.44 |
| Threshold based vs General | $H_0^1$ | 7.59 | 4 | 0.11 |
| Within general | $H_0^2$ | 2.18 | 4 | 0.70 |
| Within threshold based | | 6.44 | 8 | 0.60 |
| Poster 4a vs Guided | $H_0^3$ | 3.91 | 4 | 0.42 |
| Poster 4b vs Poster 2 | $H_0^4$ | 2.52 | 4 | 0.64. |
| *Subtable- Malawi meeting* | | | | |
| Poster Effects | | 20.02 | 16 | 0.22 |
| Threshold based vs General | $H_0^1$ | 5.34 | 4 | 0.25 |
| Within general | $H_0^2$ | 6.93 | 4 | 0.14 |
| Within threshold based | | 7.76 | 8 | 0.46 |
| Poster 4a vs Guided | $H_0^3$ | 4.04 | 4 | 0.40 |
| Poster 4b vs Poster2 | $H_0^4$ | 3.72 | 4 | 0.45 |

[†] Each row of this table presents a test of a null hypothesis of random association between the rows and columns of a contingency table, but the four highlighted here correspond to the prior hypotheses about differences among posters which are of primary interest.

[*] Probability of obtaining a deviance statistic this large or larger if the null hypothesis of random association of the rows and columns of the table hold.

Lastly, we computed mean ranks for the different professional group. We found strong evidence against the null hypothesis of random ranking for the nutritionist/health practitioners ($P$=0.017), and not for soil scientists ($P$=0.16) and agronomists ($P$=0.23).

Fig. 8 shows the mean rankings for the separate posters for all the respondents graphically. Poster 4b (raw probability plus pictograph) and 2 (IPCC verbal scale) had the largest mean ranks and poster 3 (kriging variance) had the least. Threshold based methods were found to be more effective at communicating uncertainty about spatial predictions of grain Se concentration.






**Table 8.** Analysis of Q3 according to location of meeting, professional group and methods the latter tested on separate location subtables.

| | Specified null hypothesis[†] | Deviance $(L^2)$ | Degrees of Freedom | $P^*$ |
|---|---|---|---|---|
| *Full contingency table analysis* | | | | |
| Full table | | 60.36 | 36 | 0.006 |
| Pooled within location of meeting | | 21.93 | 4 | 0.0002 |
| Pooled within professional group | | 10.01 | 8 | 0.26 |
| *Subtable- Ethiopia meeting* | | | | |
| Poster effects | | 16.60 | 16 | 0.41 |
| Threshold based vs General | $H_0^1$ | 13.48 | 4 | 0.009 |
| Within general | $H_0^2$ | 0.51 | 4 | 0.97 |
| Within threshold based | | 2.61 | 8 | 0.96 |
| Poster 4a vs Guided | $H_0^3$ | 2.03 | 4 | 0.73 |
| Poster 4b vs Poster 2 | $H_0^4$ | 0.58 | 4 | 0.97 |
| *Subtable- Malawi meeting* | | | | |
| Poster Effects | | 21.83 | 16 | 0.15 |
| Threshold based vs General | $H_0^1$ | 11.67 | 4 | 0.02 |
| Within general | $H_0^2$ | 4.07 | 4 | 0.40 |
| Within threshold based | | 6.09 | 8 | 0.64 |
| Poster 4a vs Guided | $H_0^3$ | 4.07 | 4 | 0.40 |
| Poster 4b vs Poster2 | $H_0^4$ | 2.03 | 4 | 0.73 |

[†] Each row of this table presents a test of a null hypothesis of random association between the rows and columns of a contingency table, but the four highlighted here correspond to the prior hypotheses about differences among posters which are of primary interest.

[*] Probability of obtaining a deviance statistic this large or larger if the null hypothesis of random association of the rows and columns of the table hold.

## 4 Discussion

In this study we tested strategies to communicate uncertain information through a systematic evaluation and comparison with distinct groups of data end-users. We found significant differences between participants' responses to the posters which em-
ployed general measures of uncertainty (kriging variance or confidence interval) and those which presented the probability that the Se concentration in grain falls below or above a threshold. The interpretative task that participants undertook was based on interpretation of the information relative to a nutritional threshold. The presentation of uncertainties in terms of probabilities framed with respect to this threshold was found more accessible by data users than the general measures of uncertainty, despite



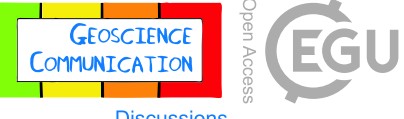

**Table 9.** Analysis of Q4 according to location of meeting, professional group and methods the latter tested on separate location subtables.

| | Specified null hypothesis[†] | Deviance $(L^2)$ | Degrees of Freedom | $P^*$ |
|---|---|---|---|---|
| *Full contingency table analysis* | | | | |
| Full table | | 25.70 | 18 | 0.11 |
| Pooled within location of meeting | | 9.14 | 2 | 0.01 |
| Pooled within professional group | | 8.96 | 4 | 0.06 |
| *Subtable- Ethiopia meeting* | | | | |
| Poster effects | | 6.47 | 8 | 0.59 |
| Threshold based vs General | $H_0^1$ | 4.34 | 2 | 0.11 |
| Within general | $H_0^2$ | 0.28 | 2 | 0.87 |
| Within threshold based | | 1.85 | 4 | 0.76 |
| Poster 4a vs Guided | $H_0^3$ | 1.22 | 2 | 0.54 |
| Poster 4b vs Poster 2 | $H_0^4$ | 0.63 | 2 | 0.73 |
| *Subtable- Malawi meeting* | | | | |
| Poster Effects | | 10.09 | 8 | 0.26 |
| Threshold based vs General | $H_0^1$ | 6.94 | 2 | 0.03 |
| Within general | $H_0^2$ | 1.61 | 2 | 0.45 |
| Within threshold based | | 1.53 | 4 | 0.82 |
| Poster 4a vs Guided | $H_0^3$ | 0.63 | 2 | 0.73 |
| Poster 4b vs Poster2 | $H_0^4$ | 0.90 | 2 | 0.64 |

[†] Each row of this table presents a test of a null hypothesis of random association between the rows and columns of a contingency table, but the four highlighted here correspond to the prior hypotheses about differences among posters which are of primary interest.

[*] Probability of obtaining a deviance statistic this large or larger if the null hypothesis of random association of the rows and columns of the table hold.

the general view (see Spiegelhalter et al., 2011) that users of information commonly find probabilities hard to interpret. Our results suggest that users of information can find information presented in terms of probabilities accessible and clear.

There was no evidence that the participants responded more positively to communication of uncertainty in the form of probabilities when these were supported with pictographs, or the calibrated phrases of the IPCC scheme, in contrast to the simple map of probability: although the maps with pictographs were highest-ranked. These methods to assist the interpretation of probability are widely used because of the assumption that many users of information find probabilities hard to interpret. However, there is evidence that calibrated phrases are themselves not without problems. Budescu et al. (2009) reported substantial inconsistencies in how people interpret scales of calibrated phrases, with a tendency to 'regressive' interpretation (interpreting large or small probabilities as close to 0.5). Jenkins et al. (2019) found that presentations of probability in numerical formats




**Table 10.** Analysis of Q5 according to location of meeting, professional group and methods the latter tested on pooled counts over Ethiopia and Malawi.

| | Specified null hypothesis[†] | Deviance $(L^2)$ | Degrees of Freedom | $P^*$ |
|---|---|---|---|---|
| *Full contingency table analysis* | | | | |
| Full table | | 40.93 | 36 | 0.26 |
| Pooled within location of meeting | | 2.55 | 4 | 0.63 |
| Pooled within professional group | | 2.35 | 8 | 0.99 |
| *Pooled counts over Ethiopia and Malawi* | | | | |
| Poster effects | | 17.74 | 16 | 0.34 |
| Threshold based vs General | $H_0^1$ | 12.23 | 4 | 0.02 |
| Within general | $H_0^2$ | 1.11 | 4 | 0.89 |
| Within threshold based | | 4.40 | 8 | 0.82 |
| Poster 4a vs Guided | $H_0^3$ | 2.34 | 4 | 0.67 |
| Poster 4b vs Poster 2 | $H_0^4$ | 2.06 | 4 | 0.72 |

[†] Each row of this table presents a test of a null hypothesis of random association between the rows and columns of a contingency table, but the four highlighted here correspond to the prior hypotheses about differences among posters which are of primary interest.

[*] Probability of obtaining a deviance statistic this large or larger if the null hypothesis of random association of the rows and columns of the table hold.

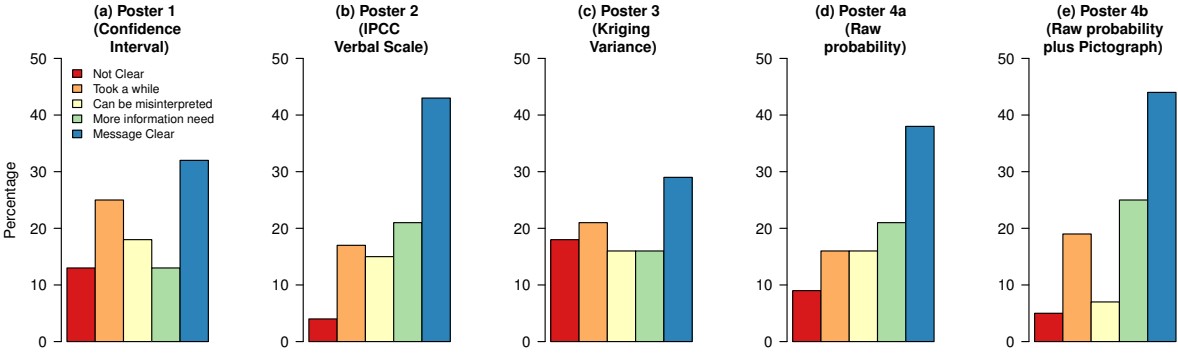

**Figure 6.** Bar charts showing how participants responded to whether a method is straightforward to interpret, Question 5.

were consistently perceived as more credible than verbal expressions. While the posters using pictographs were ranked highest (Fig. 8) in our study, we have not shown that they are markedly preferred. We note that our study focussed on stakeholders preferences and opinions, and did not include tests of how correctly the information was interpreted. We therefore suggest






**Table 11.** Analysis of Q6 according to location of meeting, professional group and methods the latter tested on pooled counts over Ethiopia and Malawi.

| | Specified null hypothesis[†] | Deviance ($L^2$) | Degrees of Freedom | $P^*$ |
|---|---|---|---|---|
| *Full contingency table analysis* | | | | |
| Full table | | 29.08 | 9 | 0.001 |
| Pooled within location of meeting | | 23.69 | 1 | 0.05 |
| Pooled within professional group | | 0.39 | 2 | 0.82 |
| *Pooled counts over Ethiopia and Malawi* | | | | |
| Poster effects | | 24.13 | 4 | <0.001 |
| Threshold based vs General | $H_0^1$ | 3.60 | 1 | <0.001 |
| Within general | $H_0^2$ | 0.002 | 1 | 0.97 |
| Within threshold based | | 0.53 | 2 | 0.77 |
| Poster 4a vs Guided | $H_0^3$ | 0.34 | 1 | 0.56 |
| Poster 4b vs Poster 2 | $H_0^4$ | 0.18 | 1 | 0.67 |

[†] Each row of this table presents a test of a null hypothesis of random association between the rows and columns of a contingency table, but the four highlighted here correspond to the prior hypotheses about differences among posters which are of primary interest.

[*] Probability of obtaining a deviance statistic this large or larger if the null hypothesis of random association of the rows and columns of the table hold.

**Table 12.** Responses to Q5 pooled counts over Ethiopia and Malawi meetings.

| Response | Pooled counts |
|---|---|
| Not clear | 27 |
| Took a while | 55 |
| Can be misinterpreted | 40 |
| More information needed | 53 |
| Message clear | 103 |

that further work is needed before a definitive assessment can be made of the value of calibrated phrases or pictographs to supplement raw probability, while noting that we have not found them to be markedly more congenial to the user.

Kriging variances were the lowest-ranked poster in the participants' overall assessment (Fig. 8). The kriging variance is fundamental to the geostatistical approach to predicting spatial variables. It is the quantity which is minimized by the kriging predictor, and its virtues as a 'built-in' measure of the uncertainty of point predictions have been widely acknowledged. None the less, it is clear that the kriging variance in itself is not an accessible measure of uncertainty for most end-users. Along with confidence intervals, the kriging variance is a general measure of uncertainty which reflects the spatial variability of the





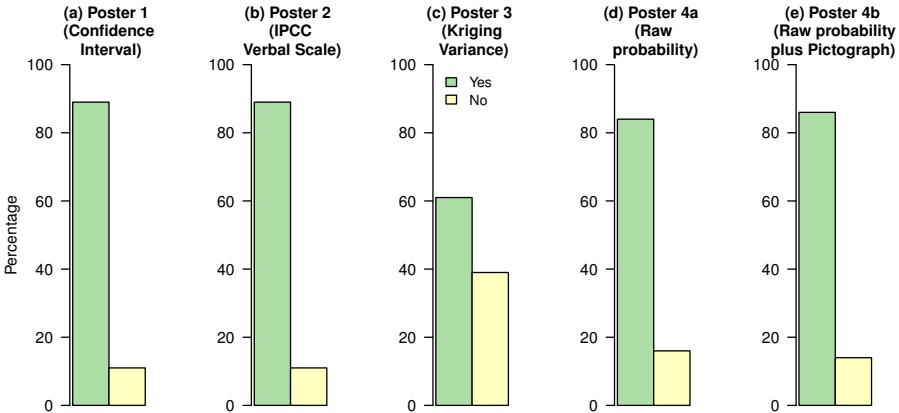

**Figure 7.** Bar charts showing how participants responded to how each poster helped them understand uncertainty in the spatial predictions, Question 6.

**Table 13.** Analysis of Q7 according to all respondents, location of meeting and professional group.

|  | Test Statistic $(X^2)$ | Degrees of Freedom | $P^*$ |
|---|---|---|---|
| All respondents | 16.90 | 4 | 0.002 |
| *Location of meeting* |  |  |  |
| Ethiopia | 7.44 | 4 | 0.12 |
| Malawi | 18.21 | 4 | 0.001 |
| *Professional group* |  |  |  |
| Agronomist | 5.60 | 4 | 0.23 |
| Soil Scientist | 6.51 | 4 | 0.16 |
| Nutritionist/Health Practitioner | 12.10 | 4 | 0.017 |

$^*$ Probability of obtaining a deviance statistic this large or larger if the null hypothesis of random ranking of the rows and columns of the table hold.

target variable and the local density of sampling. However, the user must interpret this quantity along with other information (for example, is the predicted value close to the threshold or substantially different from it) in order to make a judgement at a particular location. The probability, tied directly to the interpretative task is clearer to the user.

Confidence intervals were not ranked highly by our participants, and we had no evidence that they were found any clearer than the kriging variance. In part this might be because of the limitations of presenting the predictions and upper and lower bounds of the confidence interval as three separate maps. The task of interpreting the information at one location, or comparing






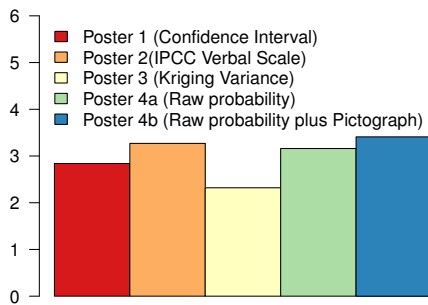

**Figure 8.** Ranking of poster in terms of the most effective at communicating uncertainty about spatial predictions.

two, when this entails examining three maps may have influenced the participants responses. In other settings the confidence

interval might be more effective for interpretation, for example where the user of information can display the confidence interval for a prediction at a site of interest as a single figure (e.g. a bar against a scale) with the threshold value of concern indicated. Further work is needed on different ways to present the confidence interval for interpretative tasks.

We only found strong evidence of differences between the meeting location for questions on interpretative and decision-focused tasks. This can be attributed to the composition of each group. Participants at the Malawi meeting comprised re-

searchers and stakeholders already somewhat engaged with the GeoNutrition project, whereas those in Ethiopia were mainly local stakeholders not previously involved with the project.

The participant groups from the two locations differed in their self-assessed level of mathematical education and use of mathematics and statistics in their work. We had more participants with mathematical components in their education up to degree level in Malawi than in Ethiopia. We had fewer people who had mathematical education only to secondary/ high school

level in Malawi than in Ethiopia. There were fewer participants who used mathematics and statistics regularly in the Ethiopia meeting. This, along with the differences in role noted in the previous paragraph, might contribute to differences between the locations. However, our data cannot support more detailed assessment of the effects of mathematical background because they are strongly unbalanced. For example we only had 3% of participants educated up to certificate/diploma level in the Ethiopia meeting. Further work on this question would require an experimental design which ensured sufficient numbers of participants

with different mathematical background.

No map is perfect (Heuvelink, 2018), but maps must be used as a basis for decisions. It is therefore important to ensure that the user of spatial information is aware of the uncertainty in these predictions, and that these are communicated in a clear way. The user must be aware that the predictions have an attached uncertainty and it is therefore possible that a decision they make might be judged incorrect in the light of perfect information. Given this, the user must have a clear enough understanding of

the uncertainty attached to a prediction so as to be confident that the decision they make will be robust given the uncertainty.





For example, the predicted concentration of a nutrient in a staple crop at a location may be such that intake of the nutrient should be sufficient to meet the needs of those who eat that crop. The user should consider the uncertainty in that prediction. If the probability that the threshold concentration is exceeded is just 0.6 (about as likely as not on the IPCC scale), then they may conclude that a decision on whether or not to proceed with an intervention at that location requires further information. If, on the other hand, the probability is 0.95 (very likely) then they may be confident in deciding to prioritize interventions elsewhere. However, if the uncertainty is not communicated clearly, then the data user might be over-confident in predictions where the probability that the threshold is exceeded is only just over 0.5, and may waste resources in further investigation or unnecessary interventions at locations where the prediction was well-supported and indicated adequate local concentrations of the nutrient.

## 5   Conclusions

Despite the general expectation that users of spatial information do not generally find probabilities a congenial way to express uncertainty, we found that when probability is used to quantify the uncertainty in a specific interpretation of spatial information, based on a nutritionally-significant threshold, end-users largely found the approach clear, and preferable to general measures of uncertainty which are not directly linked to the specific interpretation (confidence intervals and kriging variance). In the general assessment and ranking of how methods to present uncertainty succeeded, the methods based on a specific interpretation of the information, using probability, were again preferred. There was no significant evidence for a difference in assessment by users of presentations which used probability alone, and those which used pictographs or verbal phrases to aid the interpretation of the raw probability values, although these latter methods were ranked highest among all methods.

To conclude, we suggest that the challenge of communicating the significance of uncertain information to a range of stakeholders should be considered in the context of specific interpretations of the information (e.g. nutrient concentrations relative to thresholds of nutritional significance) and that, in this setting, probabilities can be accessible to a wide range of end-users. Calibrated phrases or pictographs seem to have some value (given the rankings by our participants) although there is no strong evidence that they should be preferred to a simple map of the probability. While general measures of uncertainty (kriging variance and confidence intervals) are valid ways to quantify uncertainty, they are less effective for communication, although other ways to present confidence intervals for spatial data in interactive formats online or in a GIS may merit further investigation.

*Author contributions.*   All authors contributed to the preparation of the article. CC: Conceptualisation, Methodology, Software, Formal Analysis, Investigation, Visualisation, Writing- Original Draft, Writing- Review & editing, Data Curation. JGC: Writing- Review & editing, Supervision. DG: Writing- Review & editing, Supervision, Investigation, Project administration, Funding acquisition. PCN: Writing- Original Draft, Writing- Review & editing, Supervision, Project administration, Funding acquisition. MRB: Writing- Review & editing, Supervision, Project administration, Funding acquisition. AEM: Conceptualisation, Methodology, Software, Formal Analysis, Investigation, Visualisation, Writing- Original Draft, Writing- Review & editing, Supervision, Funding acquisition. RML: Conceptualisation, Methodology, Software, Formal Analysis, Investigation, Visualisation, Writing- Original Draft, Writing- Review & editing, Supervision, Funding acquisition.



*Competing interests.* The authors declare that they have no conflict of interest.

*Acknowledgements.* This work was supported by GeoNutrition projects funded by the Bill & Melinda Gates Foundation (BMGF) [INV-009129], and the Nottingham-Rothamsted Future Food Beacon Studentships in International Agricultural Development. The funders were
not involved in the study design, the collection, management, analysis, and interpretation of data, the writing of the report or the decision to submit the report for publication.

The authors gratefully acknowledge the contributions made to this research by the participating farmers and field sampling teams. In Ethiopia, field sampling teams were from the Amhara National Regional Bureau of Agriculture. In Malawi, field sampling teams were from the Department of Agricultural Research Services, and Lilongwe University of Agriculture and Natural Resources.
We also would like to acknowledge the contributions made by E. Louise Ander, Edward JM Joy and Alexander A Kalimbira during the design of the experiment. We also appreciate Diriba B Kumssa, Adamu Belay, Mesfin Kebede, Demeke Teklu Senbetu, Kidist Gemechu and Hasset Tamirat for their roles as facilitators during the Elicitation in Malawi and Ethiopia.

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
