# Peer review of "Communicating uncertainties in spatial predictions of grain micronutrient concentration"

_Geoscience Communication, 2020_

## Referee Comment (RC1) · Anonymous Referee #1 · 17 Dec 2020

Review for Geoscience Communication gc-2020-42 (https://doi.org/10.5194/gc-2020-42)

This is a well-written manuscript on an interesting topic. My main concern is that Sections 2.3 and 3 go in very much detail on the statistical analysis. This part is difficult for an audience that has a not so strong background in statistics and it distracts the reader from the main topic of the paper: how can we communicate uncertainty about spatial predictions effectively? I strongly recommend to move large parts of these sections, including quite a few of the tables, to the Supplementary Information. Instead, more attention could be paid to what we learn from the experiment conducted on communicating uncertainty (i.e., Table 12, Figures 7 and 8). Further, a more thorough comparison should be made with findings on spatial uncertainty communication and visualisation from the cartography and geo-information literature (I added a few entry citations at the end of this review).

I also think the experiment could have been conducted in a better way and that some basic mistakes were made in preparing the posters. These and some other points are worked out in the detailed comments below. I do not require that the experiments are redone but recommendations how to do better in future could be included in the Discussion and/or Conclusion.

I should add that I did not thoroughly check the statistical methodology part, but my impression is that this is solid work, as could be expected from the author list (in particular the last author has a strong reputation in high-level statistical analysis).

Detailed comments

(L36) Not in all kriging algorithms is the prediction a linear combination of the data.

(L39, L41, L130, etc.) Authors use the term 'confidence interval', but technically this should be 'prediction interval'. There is a principal difference between the two, for example see https://en.wikipedia.org/wiki/Confidence_and_prediction_bands.

(L46, L110-114, L138) There is no need to use indicator kriging to compute exceedance probabilities. By invoking the normal distribution assumption for kriging prediction errors (which authors do, se L38), these exceedance probabilities can be easily derived from the kriging prediction and kriging variance. They will be more accurate than those obtained using indicator kriging.

(L61-65) I may be opening a box of Pandora, but authors will know that the uncertainty in the mapped concentrations of micronutrients in grain are heavily influenced by the support of the observations and predictions (i.e., the area or volume over which observations and predictions are made). Authors do not apply a change of support so the predictions and associated uncertainty refer to the support of the observations. Is this appropriate? What was it? This is not explained in L82-96: there is a lot of attention for the spatial sampling design but we learn nothing about how the field sampling was done. Were these point samples or bulk/composite samples? This is of key importance when addressing uncertainty.

(L89) was --> were.

(L103) This implies that predictions need to be back-transformed. How was this done (note that a naive back-transform returns the median, not the mean)? Information about the back-transform should be added.

(Eq. 1, L123) Here it should be upper case Z instead of lower case z, while in L132 and L134 it should be lower case z instead of upper case Z.

(L129, L340, Figure S3) Poster 3 should have shown the kriging standard deviation instead of the kriging variance. The kriging variance has different measurement units (the square of microgram per kilogram) and one cannot expect decision makers to account for this. Poster 3 also does not list the measurement units of the kriging variance. Moreover, the numbers are extremely small (around 1) and are almost certainly incorrect.

(Section 2.1.4, Figures S2 and S4) I doubt that computing the probability that the true value exceeds or lies below a threshold quantifies the uncertainty of predictions. For example, if the threshold is 38, the kriging prediction is 55 and the kriging standard deviation 8 then the probability of exceeding the threshold is extremely large (suggesting very small uncertainty, category "virtually certain"), while a kriging prediction of 36 with standard deviation 3 leads to large uncertainty (we end up in the category "about as likely as not"). But 8 is larger than 3, so can we maintain that the uncertainty of the predictions is quantified? These complications should have been addressed.

(L174) were --> where; where --> were.

(L186) Visiting posters in randomised order does not avoid carry-over effects, it only makes sure that the effects cancel out over a larger group. Perhaps rephrase this sentence to make this clear. Note also that instead of randomising it would have been better to have a deterministically determined sequence that guarantees that all posters occur in a completely balanced order.

(L207) Symbol $o_{i,j}$ not defined in the main text.

(L225) Two times "between the".

(L229, L230, L235, L296, L301) "was conducted", "participants are drawn", "This gives us", "there was", "There is". Please check entire manuscript on correct use of present and past tense.

(L277) less --> fewer.

(L324) a there is --> there is a.

(L334-335) Can and do you explain why the p-values were so different between Ethiopia and Malawi?

(Figure S1) Poster 1 has some important deficiencies. First, the mean has a continuous legend while the lower and upper limits have discrete units. This affects the map (discrete colour jumps in the limit maps). Second, all three maps should have had the same colour legend. For an example, see Figure 7 in https://onlinelibrary.wiley.com/doi/full/10.1111/ejss.12998.

*Some publications on spatial uncertainty visualisation and communications that may be used as entry to a comprehensive literature search:*

Aerts, J.C..J.H., K.C. Clarke and A.D. Keuper (2003), Testing popular visualization techniques for representing model uncertainty. Cartography and Geographic Information Science 30, 249-261.

Beven, K. R. Lamb, D. Leedal and N. Hunter (2015). Communicating uncertainty in flood inundation mapping: a case study. International Journal of River Basin Management 13, 285-295.

Johnson, C.R. and A.ER. Sanderson (2003), A next step: Visualizing errors and uncertainty. IEEE Computer Graphics and Applications 23, 6-10.

Kinkeldey, C., A.M. MacEachren and J. Schiewe (2014). How to Assess Visual Communication of Uncertainty? A Systematic Review of Geospatial Uncertainty Visualisation User Studies. Cartographic Journal 51, 372-386.

Kinkeldey, C., A.M. MacEachren, M. Riveiro and J. Schiewe (2017). Evaluating the effect of visually represented geodata uncertainty on decision-making: systematic review, lessons learned, and recommendations. Cartography and Geographic Information Science 44, 1-21.

Kunz, M., A. Gret-Regamey and L. Hurni (2011), Visualization of uncertainty in natural hazards assessments using an interactive cartographic information system. Natural Hazards 59, 1735-1751.

---

## Author Comment (AC1) · 22 Jan 2021

**Referee Comments for Geoscience Communication gc-2020-42**

Thank you for the opportunity to revise our manuscript. We would like to thank the referee for their time and for the constructive comments they have provided. We have revised the manuscript based on your suggestions. We reply to each of the comments below. Our suggested edits in the paper are in blue below, with line numbers indicating where we wish to make the changes.

| Referee Comment | Author Response |
|---|---|
| My main concern is that Sections 2.3 and 3 go in very much detail on the statistical analysis. This part is difficult for an audience that has a not so strong background in statistics and it distracts the reader from the main topic of the paper: how can we communicate uncertainty about spatial predictions effectively? I strongly recommend to move large parts of these sections, including quite a few of the tables, to the Supplementary Information. Instead, more attention could be paid to what we learn from the experiment conducted on communicating uncertainty (i.e., Table 12, Figures 7 and 8). Further, a more thorough comparison should be made with findings on spatial uncertainty communication and visualisation from the cartography and geo-information literature (I added a few entry citations at the end of this review). | Thank you for the suggestions. To address the concerns our referee on sections 2.3 and 3 in the manuscript, we have expanded the Supplementary Materials section and added an Appendix section on the manuscript. We made the following changes.

1. On L197 we changed the text to
The 'full table' illustrated in Fig. A1 is an example of this.

2. We have moved the text from L211 to L222 to the Appendix section. We also moved Fig. 2 to the Appendix and renamed it Fig. A1.

3. Fig. 3 has been moved to supplementary information of the manuscript and is now Fig. S9.

4. Tables 4 & 5 have been moved to Appendix and renamed to Tables A1 & A2. The text on L284 has been edited to
The full tables for responses for responses both locations and all posters to question Q1 are shown in Table A1 in the Appendix. The responses pooled for both meeting locations are shown in Table A2. |

| | 5. Table 12 has been moved to the Appendix and has been renamed to Table A3. The text on L319 has been edited to |
|---|---|
| | The responses for Q5 are shown in Table A3. |
| | We acknowledge, the importance of the topic raised by the referee of comparing findings on spatial uncertainty communication and visualisation from cartography. We have added a paragraph in the discussion from L404: |
| | The findings of this study complement work that has been done on cartography and visualization for spatial information (Kunz et al., 2011; Beven et al., 2015). Our findings show the importance of finding cartographic solutions to represent probability information, and to develop interactive methods for interpretation in a GIS environment (e.g., to produce pictographs, like those we have used, for sites of interest, or to find more effective ways to represent the 95% prediction interval). |
| I also think the experiment could have been conducted in a better way and that some basic mistakes were made in preparing the posters. These and some other points are worked out in the detailed comments below. I do not require that the experiments are redone but recommendations how to do better in future could be included in the Discussion and/or Conclusion | Thank for citing this, we wish to address this comment by adding a paragraph, in the discussion section, focusing on the limitations of the study. We added a paragraph in the discussion to address the concerns of the referee from L404: |
| | It is good practice to use a consistent colour scale for the three legends showing lower and upper 95% prediction interval and the conditional median. However, in our study we could not use one colour legend for the three maps for Fig. S1 (Poster 1) because of the marked differences in the predicted values on back-transformation. This made it difficult to find a working |

| | colour scale from the minimum value in the lower bound to the maximum in the upper bound on which one would see the variation in all three maps. We opted to use a continuous legend on the map of the mean and discrete ones for the lower and upper limits. This might have hindered interpretation. However, we suspect that there is a need for fundamentally different ways to visualize confidence intervals, perhaps using interactive methods to display them in a GIS environment. |
|---|---|
| (L36) Not in all kriging algorithms is the prediction a linear combination of the data. | The reviewer is correct here in that, in some circumstances, the kriging prediction may be a linear combination of some non-linear function of the data (see, for example, Webster and Oliver, 2007). It remains, however, a linear model in the parameters, hence the term "Best Linear Unbiased Predictor" for the prediction from a Linear Mixed Model. We edit the text at L36:

The prediction is a linear combination of the data, sometimes after a non-linear transformation, which is optimal. |
| (L39, L41, L130, etc.) Authors use the term 'confidence interval', but technically this should be 'prediction interval'. There is a principal difference between the two, for example see https://en.wikipedia.org/wiki/Confidence_and_prediction_bands. | Thank you for raising this suggestion, we wish to address this comment by replacing 'confidence interval' with 'prediction interval' at L11, L39, L41, L130, L134, L135, Table 1, L245, L 252, L296, L327, L345, L367, L371, L373, L374, L375, L377, and L408. We wish to make this change also on Table 1; Figures 4, 5, 6, 7 and 8. The change will also be applied to the Figure S1, S9 and S10 in the supplementary material. |
| (L46, L110-114, L138) There is no need to use indicator kriging to compute exceedance probabilities. By invoking the normal distribution assumption for kriging prediction errors (which authors do, se L38), these exceedance probabilities can be | While it certainly is possible to compute probabilities on the assumption that ordinary kriging errors are normally distributed, this does introduce an additional potential source of error. This is why methods such as indicator and |

| | |
|---|---|
| easily derived from the kriging prediction and kriging variance. They will be more accurate than those obtained using indicator kriging. | disjunctive kriging have been developed. We therefore do not accept the reviewer's view that indicator kriging would necessarily produce less accurate results than the assumption of normal errors. At line L111 we inserted the following text.

While exceedance probabilities could be computed on the assumption of normally distributed errors, we chose to use the widely-applied non-parametric method, indicator kriging, which requires no such assumption. |
| (L61-65) I may be opening a box of Pandora, but authors will know that the uncertainty in the mapped concentrations of micronutrients in grain are heavily influenced by the support of the observations and predictions (i.e., the area or volume over which observations and predictions are made). Authors do not apply a change of support so the predictions and associated uncertainty refer to the support of the observations. Is this appropriate? What was it? This is not explained in L82-96: there is a lot of attention for the spatial sampling design but we learn nothing about how the field sampling was done. Were these point samples or bulk/composite samples? This is of key importance when addressing uncertainty. | This is a fair point. The sampling method is described in detail elsewhere (Gashu et al., 2020). We have added further information about sampling in the following text we have added below L65.

The sample support for these data consisted of a bulk grain sample formed from aliquots collected from grain samples within a single field, as described by Gashu et al. (2020). The predictions, and quantifications of uncertainty, therefore, relate to grain nutrient concentrations at individual field scale. This is appropriate when considering possible health implications for smallholder and subsistence producers. |
| (L89) was --> were. | Suggested edit on L89 has been made to the manuscript.

In total, 455 sampling points were obtained, including 136 and 113 locations where teff and wheat were sampled, respectively |
| (L103) This implies that predictions need to be back-transformed. How was this done (note that a naive back- | This is also an important point. The back-transformation, to be unbiased, requires a term in the kriging variance. However, this introduces a potential source of uncertainty. For this |

| | |
|---|---|
| transform returns the median, not the mean)? Information about the back-transform should be added. | reason it is commonly advocated (e.g. Pawlowsky-Glahn and Olea (2004). Geostatistical analysis of compositional data, Oxford University Press) that the simple back-transformation by exponentiation is used. This is median-unbiased (i.e. estimates the conditional median). Pawlowsky-Glahn and Olea (2004) note that this is a more useful predictor than the conditional mean for a strongly skewed variable. We propose to expand the text at L103 to explain this, and to use the term "conditional median" rather than "conditional mean". Note, however, that the prediction interval retains its usual interpretation on back-transformation. |
| (Eq. 1, L123) Here it should be upper case Z instead of lower case z, while in L132 and L134 it should be lower case z instead of upper case Z. | An upper-case Z is used to refer to the random variable, and a lower-case z to refer to a realization. We follow sources such as Webster and Oliver (2007). We do not think that it makes a difference whether an upper or lower-case z is used for the first term in the bracket in Equation 1. We are willing to make that change at the reviewer's suggestion. However, the cases should remain unchanged at lines L132 and L134 because there we are referring to observed kriging errors (132) and are retaining the same notation for the kriging prediction (upper case) as in Equation 1. |
| (L129, L340, Figure S3) Poster 3 should have shown the kriging standard deviation instead of the kriging variance. The kriging variance has different measurement units (the square of microgram per kilogram) and one cannot expect decision makers to account for this. Poster 3 also does not list the measurement units of the kriging variance. Moreover, the numbers are extremely small (around 1) and are almost certainly incorrect. | In this study we were explicitly considering the kriging variance as a measure of prediction uncertainty, just as one might use the variance as a measure of variability. In this case we cannot back-transform the variance (or by extension the standard error) to the original units of measurement, so the kriging variance is simply presented as a relative measure of uncertainty across the mapped area. This may well be one of its disadvantages. We are not sure why the reviewer thinks the kriging variances are incorrect, we did check them by cross- |

| | |
|---|---|
| | validation. Perhaps they did not realize that these are on the log scale. At L129 we add the following text.

The kriging variance is on the transformed (log) scale, as a back-transformation of this quantity is not possible. The variations in kriging variance therefore give the interpreter an impression of the variations in prediction uncertainty across the mapped area, but not in interpretable units.

We added a comment about this in the discussion section on L368.

The difficulty of interpreting the kriging variance is compounded when a transformation is necessary, and that, in other circumstances, the kriging standard error, on the original units of measurement, may be more interpretable. |
| (Section 2.1.4, Figures S2 and S4) I doubt that computing the probability that the true value exceeds or lies below a threshold quantifies the uncertainty of predictions. For example, if the threshold is 38, the kriging prediction is 55 and the kriging standard deviation 8 then the probability of exceeding the threshold is extremely large (suggesting very small uncertainty, category "virtually certain"), while a kriging prediction of 36 with standard deviation 3 leads to large uncertainty (we end up in the category "about as likely as not"). But 8 is larger than 3, so can we maintain that the uncertainty of the predictions is quantified? These complications should have been addressed. | The reviewer makes an important point, but we do not agree that probabilities are not communicating uncertainty in these circumstances. If the prediction distribution has a large variance, but the mean is well above the threshold, then, from the perspective of a data user making a decision about nutritional interventions, the uncertainty about the contribution from staple crops is indeed small, and smaller than for a second case where the prediction variance is smaller, but the mean is near or on the threshold. |
| (L174) were --> where; where --> were. | Suggested edit on L174 has been made on the manuscript. |

| | |
|---|---|
| | Evaluation of communication methods were done through a questionnaire, as shown in Table 3, but without putting the participants in a situation where they felt they were being tested on their mathematical skills and understanding. |
| (L186) Visiting posters in randomised order does not avoid carry-over effects, it only makes sure that the effects cancel out over a larger group. Perhaps rephrase this sentence to make this clear. Note also that instead of randomising it would have been better to have a deterministically determined sequence that guarantees that all posters occur in a completely balanced order. | In view of this, we rephrased L186 to say

To avoid any bias resulting from carry-over effects from one poster to another when the individual responses were pooled for analysis.

Regarding the second point, this would still be done by randomization (e.g., a behavioural Latin square), but was not done for logistical reasons (i.e., to reduce the overall numbers of groups of participants that had to be managed in the exercise). |
| (L207) Symbol $o_{i,j}$ not defined in the main text. | The symbol $o_{i,j}$ was defined on L205 in the following way

The evidence for the saturated model, as a better model for the data than the additive model, is provided by the likelihood ratio statistic or deviance for the two models, L, where

$$L = \sum_{i=1} \sum_{j=1} o_{i,j} \ \log \frac{o_{i,j}}{e_{i,j}}$$

and $o_{i,j}$ are the number of observed response in cell [$i,j$]. |
| (L225) Two times "between the". | Suggested edit on L225 was made on the manuscript. |

| | However, it was first necessary to consider whether there was evidence for differences in the responses between the two sets of respondents at different locations. |
|---|---|
| (L229, L230, L235, L296, L301) "was conducted", "participants are drawn", "This gives us", "there was", "There is". Please check entire manuscript on correct use of present and past tense. | Thank you for this suggestion. We have checked the entire manuscript to correct on the use of present and past tense. |
| (L277) less --> fewer. | Suggested edit on L277 was made on the manuscript.

In the Ethiopia meeting, we had fewer participants (64%) who had studied mathematics and statistics up to degree level and above, than in the Malawi meeting (88%), see Fig. 3. |
| (L324) a there is --> there is a. | Suggested edit on L277 was made on the manuscript.

Fig.6 shows the responses to Q5 for the separate posters for pooled counts graphically. We can see that there is a greater proportion of respondents selecting the response `Message clear' on threshold-based methods, Posters 2 (IPCC verbal scale), 4a (raw probability) and 4b (raw probability plus pictograph), than on general based. |
| (L334-335) Can and do you explain why the p-values were so different between Ethiopia and Malawi? | The difference could be as result of differences in compositions of the groups in Ethiopia and Malawi. We added this text to explain the difference in the manuscript on L335.

The difference maybe because the stakeholder in the Malawi meeting was more homogeneous in terms of professional group (a less even distribution among them) and level of |

| | |
|---|---|
| | mathematical education than the stakeholders in the Ethiopia meeting. |
| (Figure S1) Poster 1 has some important deficiencies. First, the mean has a continuous legend while the lower and upper limits have discrete units. This affects the map (discrete colour jumps in the limit maps). Second, all three maps should have had the same colour legend. For an example, see Figure 7 in https://onlinelibrary.wiley.com/doi/full/10.1111/ejss.12998. | The reviewer makes an important point, and we must acknowledge it was difficult to find a working colour scale in which one could see the variation in all three maps, given the marked difference in the ranges. Hence, we decided to use different colours and discrete units. However, as guided by our referee, we have added paragraph explaining the limitations of the study which we wish to add from L404. |

---

## Referee Comment (RC2) · Anonymous Referee #2 · 11 Feb 2021

The paper includes a lot of statistical terminology and detail of methods. I assume intended audience is those with knowledge of statistical terminology and methods. Possible lost opportunity to appeal to a wider audience given that emphasis on communicating uncertainties.

Table 1. Would like to see the poster designs. This would add context to the subsequent discussion

Might the questions in Table 3 encourage participants to say 'Message clear' to show they understand what they are being shown? Does this introduce bias in the way the question is worded? If author agrees, there is an opportunity here to acknowledge this

[Figure]

or show has this has been accounted for in subsequent questions.

Figure 2 – Perhaps add a key to explain what the O indicates. This isn't that clear to a non-specialist

L21 – Perhaps worth alluding to the ethical issues surrounding the ethics of interventions to improve the dietary intake of Se. Whilst this is not the subject of the paper, worth noting perhaps.

L32 – Nugget variance – assumption that readers will know what this is. Author could include glossary/footnote

L225 – Good to see acknowledgement off possible differences between different groups. Suggest further group work with other participants may increase validity of study. Could this be a suggestion for future work?

L225-232 Good recognition of potential for bias

L232 Different learning styles may also affect how people interpret posters

L350 – Conclusion about users finding information presented accessible and clear – responses could have been affected by the desire to show understand the representation. I think the leading nature of the question could be seen as significant. Suggest consider acknowledging this possibility

L360-362 – Agree with statement that further work is needed

L389-390 – Agree with statement that further work needs to ensure sufficient number of participants – bigger sample size

L419-420 – Would like to see how measures of uncertainty are presented – and how these less effective methods of communication (kriging variance and confidence intervals) could be presented in a more effective way

---

## Author Comment (AC2) · 26 Feb 2021

**Referee Comments for Geoscience Communication gc-2020-42-RC2**

We would like to thank the second referee for the opportunity to revise our manuscript. We would like to thank the referee for their time and for the constructive comments they have provided. We have revised the manuscript based on your suggestions. We reply to each of the comments below. Our suggested edits in the paper are in blue below, with line numbers indicating where we wish to make the changes.

| Referee Comment | Author Response |
| --- | --- |
| The paper includes a lot of statistical terminology and detail of methods. I assume intended audience is those with knowledge of statistical terminology and methods. Possible lost opportunity to appeal to a wider audience given that emphasis on communicating uncertainties. | Thank you for the suggestions, which parallel the first comment from Referee 1. Please see our responses there. In summary, we have removed some of the statistical detail to an Appendix, including text and figures, so that the key arguments should be clearer to a general reader. |
| Table 1. Would like to see the poster designs. This would add context to the subsequent discussion | We thank the referee for raising this concern. In the manuscript we mentioned on L75 to L76 that the posters are presented in the supplementary materials. In order to make this clear for the reader we propose to add the figure number on the following lines in the manuscript:

 L129 – "To investigate the utility of the kriging variance as a method to communicate uncertainty, one poster showed a map of conditional mean of Se concentration in grain (Section 2.1.1), with a map of kriging variance (see Table 1, Fig S1)"

 L134- "One poster showed a map of conditional mean of 135 Se concentration in grain plus the lower and upper bounds of the 95% confidence intervals mapped separately to communicate the uncertainty (see Table 1, Fig S3)." |

| | L159- "Therefore, we presented three posters, each showing a map of conditional mean of Se concentration in grain (Section 2.1.1.), plus probability presented as (1) raw probability scale (see Fig S4), (2) IPCC verbal scale (see Fig S2) and (3) raw probability scale plus pictographs (see Fig S5), communicating the uncertainty (see Table 1).

. |
| Might the questions in Table 3 encourage participants to say 'Message clear' to show they understand what they are being shown? Does this introduce bias in the way the question is worded? If author agrees, there is an opportunity here to acknowledge this or show has this has been accounted for in subsequent questions. | We do not think that such a bias was likely in the context of the workshops. All responses were anonymous, and this was made very clear to participants at the start of the meeting. Furthermore (i) in the workshop we emphasized the point that the questions were not tests of the participants' understanding but rather of the efficacy of the methods for communication. (ii) It is clear in the questionnaire (and again, was emphasized in the workshop) that the participant was not being asked to interpret the representations. Rather, the interpretation was stated (e.g., "Our confidence that grain Se concentration exceeds 38 µg kg−1 is greater at x than at z") and the participant was then asked whether this was made clear by the representation. (iii) the fact that the participant was being asked to answer the same question about different methods to convey the same information emphasizes that their responses may differ between methods, even though the fixed interpretation is clear in their minds. This appears to have happened. We noted at L397 that in Malawi a large proportion of respondents selected "Not clear" as a response for the poster which used confidence intervals.

In response to comments raised by Referee 1, we proposed to include a paragraph the end of the discussion with the |

| | reflection on possible limitations of the study, from L404. We would like to add the following text to the paragraph from L404: |
|---|---|
| | "We accept that a possible source of bias in any such study is that a participant feels that they are being tested on their interpretative skills, and so might select a response which suggests, in a general sense, that they understand the input (e.g. "Message clear" for the case in Table 3). However, all participants were aware that their responses were strictly anonymous, and it was emphasized that the task involved their evaluation of several methods for the communication of an interpretation which was provided.  In future studies it might be useful to include some final questions which actually are "tests of interpretation" secondary to the main task, to see whether this affects the responses given for different methods." |
| Figure 2 – Perhaps add a key to explain what the O indicates. This isn't that clear to a non-specialist | We propose to add a key to Figure 2 as suggested (renamed as Fig A1 in the appendix section). |
| L21 – Perhaps worth alluding to the ethical issues surrounding the ethics of interventions to improve the dietary intake of Se. Whilst this is not the subject of the paper, worth noting perhaps. | This is an interesting suggestion. We do not think that the general ethics of food-based interventions is within the scope of this study. However, we propose to the following comment in the Conclusions from L412:

"Because decisions on interventions to address nutrient deficiencies may have positive and negative effects on peoples' health and well-being, the interpretation of information such as that we have used is not value-neutral, and uncertainty in information has ethical implications (given |

| | that all spatial information is uncertain, how much uncertainty is ethically acceptable in the decision process?). While these considerations are outside the scope of the study reported here, it would be interesting in future research to examine how individual attitudes to the ethics of fortification interventions affect their responses, and whether individuals' perspectives on the ethical implications of basing decisions on uncertain information differs between different methods to communicate that uncertainty." |
|---|---|
| L32 – Nugget variance – assumption that readers will know what this is. Author could include glossary/footnote | We propose to edit the sentence which starts at l30 of the paper to read:

"Predictions are subject to uncertainty because of spatial variability resulting from multiple factors operating at different scales (Lark et al., 2014). In addition to environmental factors (geology, climate), there is also uncertainty due to measurement error in the analysis of material, and sampling error in the field where a single crop or soil sample is collected." |
| L225 – Good to see acknowledgement off possible differences between different groups. Suggest further group work with other participants may increase validity of study. Could this be a suggestion for future work? | The reviewer makes an important point, and we propose the following edit to the text on L389 to emphasize this point.

Further work to address this question and examine how stakeholders interpreted each poster will require an elicitation with sufficient numbers of participants with different mathematical background. |
| L225-232 Good recognition of potential for bias | Thank you for the acknowledgement. |

| | |
|---|---|
| L232 Different learning styles may also affect how people interpret posters | We agree and therefore we expected this to affect their responses. However, due to unbalanced numbers of participants when we categorised them by level of mathematical education, it was not possible to do further analysis and we propose editing the text on this at L389 to read:

"Further work on this question would require an experimental design which ensured sufficient numbers of participants with different mathematical backgrounds.  This would be useful to understand better how different learning styles influence the interpretation of uncertain information".
. |
| L350 – Conclusion about users finding information presented accessible and clear – responses could have been affected by the desire to show understand the representation. I think the leading nature of the question could be seen as significant. Suggest consider acknowledging this possibility | Please see our response to the third point above. We do not agree that the participants were asked a leading question. They were asked to select among responses to a question about whether it was clear from the poster that a certain statement was true, and possible responses included "Not clear" and "More information needed" as well as "Message clear". |
| L360-362 – Agree with statement that further work is needed | Thank you for the acknowledgement. |
| L419-420 – Would like to see how measures of uncertainty are presented – and how These less effective methods of communication (kriging variance and confidence intervals) could be presented in a more effective way | Thank you for acknowledging this point and we strongly believe this is a scope for future research work on methods of communicating uncertainties in spatial predictions. |

---

## Author Response (AR1)

**Response to Referee Comments for gc-2020-42**

We would like to thank the referees for the opportunity to revise our manuscript. We have revised the manuscript based these suggestions and the changes are shown in the tracked changed version of the manuscript. The line numbers which we refer to are the ones in the tracked change manuscript.

**Referee 1**

| Referee Comment | Author Response |
|---|---|
| My main concern is that Sections 2.3 and 3 go in very much detail on the statistical analysis. This part is difficult for an audience that has a not so strong background in statistics and it distracts the reader from the main topic of the paper: how can we communicate uncertainty about spatial predictions effectively? I strongly recommend to move large parts of these sections, including quite a few of the tables, to the Supplementary Information. Instead, more attention could be paid to what we learn from the experiment conducted on communicating uncertainty (i.e., Table 12, Figures 7 and 8).

 Further, a more thorough comparison should be made with findings on spatial uncertainty communication and visualisation from the cartography and geo-information literature (I added a few entry citations at the end of this review). | Thank you for the suggestions. To address the concerns our referee on sections 2.3 and 3 in the manuscript, we have expanded the Supplementary Materials section and added an Appendix section on the manuscript. We made the following changes on the manuscript.

 1. We have moved the text from L242 to L253 to the Appendix section (L510 to L524).
 2. We also moved Fig. 2 to the Appendix and renamed it Fig. A1
 3. Fig. 3 has been moved to supplementary information of the manuscript and is now Fig. S9.

 4. Tables 4 & 5 have been moved to Appendix and renamed to Tables A1 & A2.

 5. The text on L316 to L317 has been edited to
 The full tables for responses for responses both locations and all posters to question Q1 are shown in Table A1 in the Appendix. The responses pooled for both meeting locations are shown in Table A2. |

| | |
|---|---|
| | 6. Table 12 has been moved to the Appendix and has been renamed to Table A3.
7. The text on L370 has been edited to
The responses for Q5 are shown in Table A3.

We acknowledge, the importance of the topic raised by the referee of comparing findings on spatial uncertainty communication and visualisation from cartography. We have added a paragraph in the discussion from L469 to L472:

The findings of this study complement work that has been done on cartography and visualization for spatial information (Kunz et al., 2011; Beven et al., 2015). Our findings show the importance of finding cartographic solutions to represent probability information, and to develop interactive methods for interpretation in a GIS environment (e.g., to produce pictographs, like those we have used, for sites of interest, or to find more effective ways to represent the 95% prediction interval). |
| I also think the experiment could have been conducted in a better way and that some basic mistakes were made in preparing the posters. These and some other points are worked out in the detailed comments below. I do not require that the experiments are redone but recommendations how to do better in future could be included in the Discussion and/or Conclusion | Thank for citing this, we wish to address this comment by adding a paragraph, in the discussion section, focusing on the limitations of the study. We added a paragraph in the discussion to address the concerns of the referee from L472 to L479.

It is good practice to use a consistent colour scale for the three legends showing lower and upper 95% prediction interval and the conditional median. However, in our study we could not use one colour legend for the three maps for Fig. S1 (Poster 1) because of the marked differences in the predicted values |

| | on back-transformation. This made it difficult to find a working colour scale from the minimum value in the lower bound to the maximum in the upper bound on which one would see the variation in all three maps. We opted to use a continuous legend on the map of the mean and discrete ones for the lower and upper limits. This might have hindered interpretation. However, we suspect that there is a need for fundamentally different ways to visualize confidence intervals, perhaps using interactive methods to display them in a GIS environment. |
|---|---|
| (L36) Not in all kriging algorithms is the prediction a linear combination of the data. | The referee is correct here in that, in some circumstances, the kriging prediction may be a linear combination of some non-linear function of the data (see, for example, Webster and Oliver, 2007). It remains, however, a linear model in the parameters, hence the term "Best Linear Unbiased Predictor" for the prediction from a Linear Mixed Model. We edited the text at L37:

 The prediction is a linear combination of the data, sometimes after a non-linear transformation, which is optimal. |
| (L39, L41, L130, etc.) Authors use the term 'confidence interval', but technically this should be 'prediction interval'. There is a principal difference between the two, for example see https://en.wikipedia.org/wiki/Confidence_and_prediction_bands. | We agree with the referee and we have replaced 'confidence interval' with 'prediction interval' at L11, L46, L153, L158, L159, L279, L283, L378, L405, L428, L434, L436, L438, L439, L441, L489, L491, L508 and L509.

 We also have made this change on Table 1; Figures 2, 3, 4, 5 and 6 in the manuscript. The change will also be applied to the Figure S1, S10 and S11 in the supplementary material. |

| | |
|---|---|
| (L46, L110-114, L138) There is no need to use indicator kriging to compute exceedance probabilities. By invoking the normal distribution assumption for kriging prediction errors (which authors do, se L38), these exceedance probabilities can be easily derived from the kriging prediction and kriging variance. They will be more accurate than those obtained using indicator kriging. | While it certainly is possible to compute probabilities on the assumption that ordinary kriging errors are normally distributed, this does introduce an additional potential source of error. This is why methods such as indicator and disjunctive kriging have been developed. We therefore do not accept the reviewer's view that indicator kriging would necessarily produce less accurate results than the assumption of normal errors. At L129 to L130 we have inserted the following text.

While exceedance probabilities could be computed on the assumption of normally distributed errors, we chose to use the widely-applied non-parametric method, indicator kriging, which requires no such assumption. |
| (L61-65) I may be opening a box of Pandora, but authors will know that the uncertainty in the mapped concentrations of micronutrients in grain are heavily influenced by the support of the observations and predictions (i.e., the area or volume over which observations and predictions are made). Authors do not apply a change of support so the predictions and associated uncertainty refer to the support of the observations. Is this appropriate? What was it? This is not explained in L82-96: there is a lot of attention for the spatial sampling design but we learn nothing about how the field sampling was done. Were these point samples or bulk/composite samples? This is of key importance when addressing uncertainty. | This is a fair point. The sampling method is described in detail elsewhere (Gashu et al., 2020). We have added further information about sampling from L95 to L98:

The sample support for these data consisted of a bulk grain sample formed from aliquots collected from grain samples within a single field, as described by Gashu et al. (2020). The predictions, and quantifications of uncertainty, therefore, relate to grain nutrient concentrations at individual field scale. This is appropriate when considering possible health implications for smallholder and subsistence producers. |
| (L89) was --> were. | The suggested edit has been made on L89: |

| | In total, 455 sampling points were obtained, including 136 and 113 locations where teff and wheat were sampled, respectively |
|---|---|
| (L103) This implies that predictions need to be back-transformed. How was this done (note that a naive back-transform returns the median, not the mean)? Information about the back-transform should be added. | This is also an important point. The back-transformation, to be unbiased, requires a term in the kriging variance. However, this introduces a potential source of uncertainty. For this reason it is commonly advocated (e.g. Pawlowsky-Glahn and Olea (2004). Geostatistical analysis of compositional data, Oxford University Press) that the simple back-transformation by exponentiation is used. This is median-unbiased (i.e. estimates the conditional median). Pawlowsky-Glahn and Olea (2004) note that this is a more useful predictor than the conditional mean for a strongly skewed variable.

We have added a paragraph from L118 to L123 to explain this, and to use the term "conditional median" rather than "conditional mean". Note, however, that the prediction interval retains its usual interpretation on back-transformation. |
| (Eq. 1, L123) Here it should be upper case Z instead of lower case z, while in L132 and L134 it should be lower case z instead of upper case Z. | An upper-case Z is used to refer to the random variable, and a lower-case z to refer to a realization. We follow sources such as Webster and Oliver (2007). We do not think that it makes a difference whether an upper or lower-case z is used for the first term in the bracket in Equation 1. We are willing to make that change at the reviewer's suggestion. However, the cases should remain unchanged at lines L139 and L140 because there we are referring to observed kriging errors (L139) and are retaining the same notation for the kriging prediction (upper case) as in Equation 1. |
| (L129, L340, Figure S3) Poster 3 should have shown the kriging standard deviation instead of the kriging variance. The kriging | In this study we were explicitly considering the kriging variance as a measure of prediction uncertainty, just as one might use |

variance has different measurement units (the square of microgram per kilogram) and one cannot expect decision makers to account for this. Poster 3 also does not list the measurement units of the kriging variance. Moreover, the numbers are extremely small (around 1) and are almost certainly incorrect.

the variance as a measure of variability. In this case we cannot back-transform the variance (or by extension the standard error) to the original units of measurement, so the kriging variance is simply presented as a relative measure of uncertainty across the mapped area. This may well be one of its disadvantages. We are not sure why the reviewer thinks the kriging variances are incorrect, we did check them by cross-validation. Perhaps they did not realize that these are on the log scale. We have expanded the text from L140 to 144 and from L149 to L151 to explain this:

The kriging variance is on the transformed (log) scale, as a back-transformation of this quantity is not possible. The variations in kriging variance therefore give the interpreter an impression of the variations in prediction uncertainty across the mapped area, but not in interpretable units.

We also have added a comment about this in the discussion section on L429 to L431.

The difficulty of interpreting the kriging variance is compounded when a transformation is necessary, and that, in other circumstances, the kriging standard error, on the original units of measurement, may be more interpretable.

| | |
|---|---|
| (Section 2.1.4, Figures S2 and S4) I doubt that computing the probability that the true value exceeds or lies below a threshold quantifies the uncertainty of predictions. For example, if the threshold is 38, the kriging prediction is 55 and the kriging standard deviation 8 then the probability of exceeding the threshold is extremely large (suggesting very small uncertainty, category "virtually certain"), while a kriging prediction of 36 with standard deviation 3 leads to large uncertainty (we end up in the category "about as likely as not"). But 8 is larger than 3, so can we maintain that the uncertainty of the predictions is quantified? These complications should have been addressed. | The reviewer makes an important point, but we do not agree that probabilities are not communicating uncertainty in these circumstances. If the prediction distribution has a large variance, but the mean is well above the threshold, then, from the perspective of a data user making a decision about nutritional interventions, the uncertainty about the contribution from staple crops is indeed small, and smaller than for a second case where the prediction variance is smaller, but the mean is near or on the threshold. |
| (L174) were --> where; where --> were. | Suggested edit from L200 to L202 has been made on the manuscript.

Evaluation of communication methods was done through a questionnaire, as shown in Table 3, but without putting the participants in a situation where they felt they were being tested on their mathematical skills and understanding. |
| (L186) Visiting posters in randomised order does not avoid carry-over effects, it only makes sure that the effects cancel out over a larger group. Perhaps rephrase this sentence to make this clear. Note also that instead of randomising it would have been better to have a deterministically determined sequence that guarantees that all posters occur in a completely balanced order. | In view of this, we rephrased the sentence from L213 to L214 to:

Participants visited each poster in a randomised order to avoid any bias resulting from carry-over effects from one poster to another when the individual responses were pooled for analysis.

Regarding the second point, this would still be done by randomization (e.g., a behavioural Latin square), but was not |

| | done for logistical reasons (i.e., to reduce the overall numbers of groups of participants that had to be managed in the exercise). |
|---|---|
| (L207) Symbol $o_{i,j}$ not defined in the main text. | The symbol $o_{i,j}$ was defined on L235 to L238 in the following way

The evidence for the saturated model, as a better model for the data than the additive model, is provided by the likelihood ratio statistic or deviance for the two models, L, where

$$L = \sum_{i=1}\sum_{j=1} o_{i,j} \; \log\frac{o_{i,j}}{e_{i,j}}$$

and $o_{i,j}$ are the number of observed responses in cell [*I,j*]. |
| (L225) Two times "between the". | Suggested edit on L255 has been made

However, it was first necessary to consider whether there was evidence for differences in the responses between the two sets of respondents at different locations. |
| (L229, L230, L235, L296, L301) "was conducted", "participants are drawn", "This gives us", "there was", "There is". Please check entire manuscript on correct use of present and past tense. | Thank you for this suggestion. We have checked the entire manuscript to correct on the use of present and past tense. |
| (L277) less --> fewer. | Suggested edit on L309 was made on the manuscript.

In the Ethiopia meeting, we had fewer participants (64%) who had studied mathematics and statistics up to degree level and above, than in the Malawi meeting (88%), see Fig S9 . |

| | |
|---|---|
| (L324) a there is --> there is a. | Suggested edit on L375 to L377 has been made on the manuscript.

Fig.4 shows the responses to Q5 for the separate posters for pooled counts graphically. We can see that there is a greater proportion of respondents selecting the response `Message clear' on threshold-based methods, Posters 2 (IPCC verbal scale), 4a (raw probability) and 4b (raw probability plus pictograph), than on general based. |
| (L334-335) Can and do you explain why the p-values were so different between Ethiopia and Malawi? | The difference could be as result of differences in compositions of the groups in Ethiopia and Malawi. We added this text to explain the difference in the manuscript on L386 to L388.

The difference maybe because the set of stakeholders in the Malawi meeting was more homogeneous in terms of professional group (a less even distribution among them) and level of mathematical education than the stakeholders in the Ethiopia meeting. |
| (Figure S1) Poster 1 has some important deficiencies. First, the mean has a continuous legend while the lower and upper limits have discrete units. This affects the map (discrete colour jumps in the limit maps). Second, all three maps should have had the same colour legend. For an example, see Figure 7 in https://onlinelibrary.wiley.com/doi/full/10.1111/ejss.12998. | The reviewer makes an important point, and we must acknowledge it was difficult to find a working colour scale in which one could see the variation in all three maps, given the marked difference in the ranges. Hence, we decided to use different colours and discrete units. However, as guided by our referee, we have added paragraph explaining the limitations of the study from L473 to L480. |

**Referee 2**

| Referee Comment | Author Response |
|---|---|
| The paper includes a lot of statistical terminology and detail of methods. I assume intended audience is those with knowledge of statistical terminology and methods. Possible lost opportunity to appeal to a wider audience given that emphasis on communicating uncertainties. | Thank you for the suggestions, which parallel the first comment from Referee 1. Please see our responses there. In summary, we have removed some of the statistical detail to an Appendix, including text and figures, so that the key arguments should be clearer to a general reader. |
| Table 1. Would like to see the poster designs. This would add context to the subsequent discussion | In the manuscript we mentioned on L81to L82 that the posters are presented in the supplementary materials. In order to make this clear for the reader we have added the figure number on the following lines in the manuscript:

L151to L152 – "To investigate the utility of the kriging variance as a method to communicate uncertainty, one poster showed a map of conditional mean of Se concentration in grain (Section 2.1.1), with a map of kriging variance (see Table 1, Fig S1)"

L157 to L159- "One poster showed a map of conditional mean of 135 Se concentration in grain plus the lower and upper bounds of the 95% confidence intervals mapped separately to communicate the uncertainty (see Table 1, Fig S3)."

L183 to L185- "Therefore, we presented three posters, each showing a map of conditional mean of Se concentration in grain (Section 2.1.1.), plus probability presented as (1) raw probability scale (see Fig S4), (2) IPCC verbal scale (see Fig |

| | S2) and (3) raw probability scale plus pictographs (see Fig S5), communicating the uncertainty (see Table 1). |
| --- | --- |
| | . |
| Might the questions in Table 3 encourage participants to say 'Message clear' to show they understand what they are being shown? Does this introduce bias in the way the question is worded? If author agrees, there is an opportunity here to acknowledge this or show has this has been accounted for in subsequent questions. | We do not think that such a bias was likely in the context of the workshops. All responses were anonymous, and this was made very clear to participants at the start of the meeting. Furthermore (i) in the workshop we emphasized the point that the questions were not tests of the participants' understanding but rather of the efficacy of the methods for communication. (ii) It is clear in the questionnaire (and again, was emphasized in the workshop) that the participant was not being asked to interpret the representations. Rather, the interpretation was stated (e.g., "Our confidence that grain Se concentration exceeds 38 µg kg−1 is greater at x than at z") and the participant was then asked whether this was made clear by the representation. (iii) the fact that the participant was being asked to answer the same question about different methods to convey the same information emphasizes that their responses may differ between methods, even though the fixed interpretation is clear in their minds. This appears to have happened. We noted at L397 that in Malawi a large proportion of respondents selected "Not clear" as a response for the poster which used confidence intervals. |
| | In response to comments raised by Referee 1, we added a paragraph the end of the discussion with the reflection on possible limitations of the study. To expand the discussion on the limitations of the study, we have added the following paragraph from L481 to L486: |

| | "We accept that a possible source of bias in any such study is that a participant feels that they are being tested on their interpretative skills, and so might select a response which suggests, in a general sense, that they understand the input (e.g. "Message clear" for the case in Table 3). However, all participants were aware that their responses were strictly anonymous, and it was emphasized that the task involved their evaluation of several methods for the communication of an interpretation which was provided. In future studies it might be useful to include some final questions which actually are "tests of interpretation" secondary to the main task, to see whether this affects the responses given for different methods." |
|---|---|
| Figure 2 – Perhaps add a key to explain what the O indicates. This isn't that clear to a non-specialist | We added the key to Fig A1 (renamed from Fig 2) as suggested by the referee. |
| L21 – Perhaps worth alluding to the ethical issues surrounding the ethics of interventions to improve the dietary intake of Se. Whilst this is not the subject of the paper, worth noting perhaps. | This is an interesting suggestion. We do not think that the general ethics of food-based interventions is within the scope of this study. However, we added the following comment in the Conclusions from L496 to L503:

"Because decisions on interventions to address nutrient deficiencies may have positive and negative effects on peoples' health and well-being, the interpretation of information such as that we have used is not value-neutral, and uncertainty in information has ethical implications (given that all spatial information is uncertain, how much uncertainty is ethically acceptable in the decision process?). While these considerations are outside the scope of the study reported here, it would be interesting in future research to examine |

| | how individual attitudes to the ethics of fortification interventions affect their responses, and whether individuals' perspectives on the ethical implications of basing decisions on uncertain information differs between different methods to communicate that uncertainty.'' |
|---|---|
| L32 – Nugget variance – assumption that readers will know what this is. Author could include glossary/footnote | We have expanded the text from L32 to L34 to explain the nugget variance.

"Predictions are subject to uncertainty because of spatial variability resulting from multiple factors operating at different scales (Lark et al., 2014). In addition to environmental factors (geology, climate), there is also uncertainty due to measurement error in the analysis of material, and sampling error in the field where a single crop or soil sample is collected." |
| L225 – Good to see acknowledgement off possible differences between different groups. Suggest further group work with other participants may increase validity of study. Could this be a suggestion for future work? | The reviewer makes an important point, and we made the following edit to the text on L453 to L456 to emphasize this point.

Further work to address this question and examine how stakeholders interpreted each poster will require an elicitation with sufficient numbers of participants with different mathematical background. |
| L225-232 Good recognition of potential for bias | Thank you for the acknowledgement. |
| L232 Different learning styles may also affect how people interpret posters | We agree and therefore we expected this to affect their responses. However, due to unbalanced numbers of participants when we categorised them by level of |

| | mathematical education, it was not possible to do further analysis. We acknowledge this and we highlighted this as a future work from L453 to L456:

Further work to address this question and examine how stakeholders interpreted each poster will require an elicitation with sufficient numbers of participants with different mathematical background. This would be useful to understand better how different learning styles influence the interpretation of uncertain information. |
|---|---|
| L350 – Conclusion about users finding information presented accessible and clear – responses could have been affected by the desire to show understand the representation. I think the leading nature of the question could be seen as significant. Suggest consider acknowledging this possibility | Please see our response to the third point above. We do not agree that the participants were asked a leading question. They were asked to select among responses to a question about whether it was clear from the poster that a certain statement was true, and possible responses included "Not clear" and "More information needed" as well as "Message clear". |
| L360-362 – Agree with statement that further work is needed | Thank you for the acknowledgement. |
| L419-420 – Would like to see how measures of uncertainty are presented – and how These less effective methods of communication (kriging variance and confidence intervals) could be presented in a more effective way | Thank you for acknowledging this point and we strongly believe this is a scope for future research work on methods of communicating uncertainties in spatial predictions. |

---

## Author Response (AR2)

**Response to Referee Comments for gc-2020-42**

We would like to thank the referee for the opportunity to revise our manuscript. We have revised the manuscript based these suggestions and the changes are shown in the tracked changed version of the manuscript. Our edits in the paper are in blue below, and the line numbers refer to the ATC version of the revision.

**Referee 1**

| Referee Comment | Author Response |
| --- | --- |
| (L39, Eq. 1) I maintain that the lower case z should be an upper case Z. Here, the random function model is invoked and the kriging variance refers to the variance of the kriging prediction error $Z(x\_0)-Z\sim(x\_0)$. I am surprised that authors (in particular the last author) maintain their earlier view that it should be $E[\{z(x\_0)-Z\sim(x\_0)\}^2]$. Authors refer to Webster and Oliver (2007) to support their claim, but please check Eq. 8.2 in their book, which clearly writes $E[\{Z(x\_0)-Z\sim(x\_0)\}^2]$, as it should be. Note that in case of $E[\{z(x\_0)-Z\sim(x\_0)\}^2]$ we have that $z(x\_0)$ is a deterministic constant and so we effectively get the variance of $Z\sim(x\_0)$, which is very different from the variance of $Z(x\_0)-Z\sim(x\_0)$. For example, suppose $x\_0$ is a measurement location. In that case we would get $Z\sim(x\_0)=Z(x\_o)$ and hence $E[\{Z(x\_0)-Z\sim(x\_0)\}^2]=0$, while $E[\{z(x\_0)-Z\sim(x\_0)\}^2]=Var(Z(x\_0))$. In line 155 it should be $z\sim(x\_0)$ instead of $Z\sim(x\_0)$, because here you refer to the cross-validation prediction errors, which are deterministic values, not random variables. | The suggest change has been made on L132 to L133, Equation 1. |
| I maintain that authors should in Poster 3 better have presented the kriging standard deviation instead of the kriging variance. From their response I now understand that the kriging variance is that of the | We do accept the severe limitations of the kriging variance as a means for communicating uncertainty, particularly here where a transformation is necessary. We are not |

log-transformed selenium content, while the kriging prediction is the back-transformed (hence median) selenium content. This explains the extremely low values. Now I wonder, does it make sense to present a kriging prediction map of the back-transformed variable and a kriging variance map of the transformed variable? How can we ever expect users to make sense of that and grasp the uncertainty? Note that the spatial pattern of the variance map is strongly influenced by whether it is done for the log-transformed or back-transformed variable. Authors justify showing a kriging variance map of the log-transformed selenium because "we cannot back-transform" the variance. Well, why not use Eq. 8.40 in the Webster and Oliver book? Admittedly this refers to simple kriging, which is not the same as ordinary kriging, but the difference will be small. All in all quite unsatisfactory how uncertainty was communicated in Poster 3: not only authors show the kriging variance instead of the kriging standard deviation, but they also mix up log-transformed and back-transformed variables. As before, I am not requesting that the questionnaires are redone, but authors should mention the weaknesses of their approach in the Discussion. The text that they now include in the revision (L148-L150, L429-431), is a good step in this direction, but they might add that they could have done better."

convinced that kriging standard error would solve the problem in the case of log-normal variables, because the units are still on the log-scale, and the prediction interval is an obvious way to turn the kriging variance into an interpretable measure on the original scales of measurement. Neither are we happy with the idea of using the simple kriging variance as the referee suggests, because this would be anticonservative. We included kriging variance among the methods of communication because, as a standard raw output from the kriging equations which does indeed characterize local prediction uncertainty, it is commonly presented in map form as a measure of the reliability of predictions. We therefore add the following text from L141 to L157 to emphasize that these limitations were clear a priori.:

Because the kriging variance is a direct output of kriging algorithms, it is common to see it mapped alongside kriging predictions and referred to as a measure of local prediction uncertainty (e.g. Holmes et al., 2007; Goovaerts, 2014; Hatvani et al., 2021). However, the interpretation of the kriging variance may be challenging, particularly for a non-specialist user of spatial information. One could take its square root, and present it as a kriging standard error with the same units as the target variable. However, the interpretation of the raw standard error can clearly be helped by rescaling it to a prediction interval, and we consider this option in the next section.

The interpretation of the kriging variance is particularly difficult in the case of a variable which must be transformed prior to analysis. The kriging variance cannot be backtransformed to the original units (except for simple kriging). In this setting then, the kriging variance can serve as little more than a general uncertainty index, indicating in general where uncertainty is large, and where it is small. However, such generalized indices have been developed for 3-D geological information to serve the needs of engineering stakeholders (e.g Lelliott et al., 2009; Lark et al., 2014). For this reason, and because of the longstanding use of kriging variance as an uncertainty measure (see above), we included it as a measure of uncertainty in this experiment. One poster showed a map of the conditional median of Se concentration in grain (Section 2.1.1), with a map of kriging variance on the transformed units (see Table 1, Fig S3).

The referee has an important point and in this context the kriging variance can serve as little more than a general "uncertainty index", and is therefore unlikely to be useful, and that the responses we received confirmed this. In cases where transformation is not an issue the kriging standard error, on the original units of measurement, may be more interpretable to the end user, but it remains an abstract quantity. Rescaling it to a confidence interval, presented either by the limits, or by its width, is likely to be more useful, although our results suggest that the communication of confidence intervals requires more attention. Although the kriging variance is a valid statistic it has very little value as a means for communicating uncertainty for a general audience. Therefore, we have

Following on from this, we expanded the text on L388 to L401 to include the suggestions made by the referee.

| | "Although the kriging variance is a valid statistic, in this context it has very little value as a means for communicating uncertainty for a general audience. That is particularly true in this case, where the kriging variance must remain on transformed units, and so serves as little more than a general ``uncertainty index". This was clear a priori, and is confirmed by the responses we received. Our findings here cannot, therefore, be regarded as definitive, and a similar experiment for variables which do not require transformation would be necessary in further research. In such cases one could also include the kriging standard error as an uncertainty measure, to assess (i) whether the fact it is presented in the units of the target variable makes it preferable to kriging variance and (ii) whether it is regarded as less-interpretable than its rescaled form as a prediction interval. That said, our results do show that the communication of prediction intervals requires more attention.

These considerations aside, kriging variances, standard error and prediction intervals must be interpreted by the user along with other information (for example, is the predicted value close to the threshold or substantially different from it) in order to make a judgement at a particular location. Our results do show that probability measured, tied directly to the interpretative task, are clearer to the user than general measures of uncertainty." |

Goovaerts, P. 2014. Geostatistics: a common link between medical geography, mathematical geology, and

medical geology. Journal of the Southern African Institute of Mining and Metallurgy, 114: 605–612

Hatvani, I.G., Szatmári, G., Kern, Z., Erdélyi, D., Vreča, P., Kanduč, T., Czuppon, G., Lojen, S., Kohán , B. 2021. Geostatistical evaluation of the design of the precipitation stable isotope monitoring network for Slovenia and Hungary. Environment International, 146, 106263.

Holmes, K.W., Van Niel, K.P., Kendrick, G.A. Radford, B. 2007. Probabilistic large-area mapping of seagrass species distributions. Aquatic Conservation: Marine and Freshwater Ecosystems, 17, 385 – 407.

Lark, R.M., Mathers, S.J., Marchant, A., Hulbert, A. 2014. An index to represent lateral variation of the confidence of experts in a 3-D geological model. Proceedings of the Geologists Association, 125, 267–278

Lelliott, M.R., Cave, M.R., Wealthall, G.P., 2009. A structured approach to the measurement of uncertainty in 3D geological models. Quarterly Journal of Engineering Geology and Hydrogeology. 42, 95–105.